# Recurrent neural networks: vanishing and exploding gradients are not the end of the story

**Nicolas Zucchet**
Department of Computer Science
ETH Zürich
nzucchet@ethz.ch

**Antonio Orvieto**
ELLIS Institute Tübingen
MPI for Intelligent Systems
Tübingen AI Center
antonio@tue.ellis.eu

## Abstract

Recurrent neural networks (RNNs) notoriously struggle to learn long-term memories, primarily due to vanishing and exploding gradients. The recent success of deep state-space models (SSMs), a subclass of RNNs, to overcome such difficulties challenges our theoretical understanding. In this paper, we delve into the optimization challenges of RNNs and discover that, as the memory of a network increases, changes in its parameters result in increasingly large output variations, making gradient-based learning highly sensitive, even without exploding gradients. Our analysis further reveals the importance of the element-wise recurrence design pattern combined with careful parametrizations in mitigating this effect. This feature is present in deep SSMs, as well as in other architectures, such as LSTMs. Overall, our insights provide a new explanation for some of the difficulties in gradient-based learning of RNNs and why some architectures perform better than others.

Recurrent neural networks [RNNs; 1, 2] have long been the canonical architecture for modeling temporal data [3, 4]. However, they are notoriously difficult to train on long sequences, as error signals flowing backward in time tend to either vanish or explode [5–8]. Attention mechanisms [9], as featured in transformers [10], address these issues by enabling direct token-to-token communication, considerably simplifying signal propagation across long time intervals. Yet, their superior performance comes with increased computational and memory costs, due to their quadratic scaling in the sequence length. This limitation has motivated significant research aimed at making transformers more efficient [11–15].

A promising line of research in this direction involves a new type of linear recurrent networks known as deep state-space models [SSMs; 16–22]. These models trade expressivity for faster training speed, and they have been shown to be particularly effective at capturing long-range dependencies. In this paper, we wonder whether this effectiveness can be solely attributed to their ability to avoid vanishing and exploding gradients. The simplicity of such models presents an opportunity for in-depth theoretical analysis. We focus on signal propagation within these models.

After reviewing classical results on recurrent neural networks in Section 1, we demonstrate that they can suffer from an understudied problem: as the recurrent network encodes longer memories, the network's activity becomes increasingly sensitive to changes in its parameters, even when its dynamics remains stable. In Section 3, we then show that SSMs, as well as other architectures such as LSTMs, are well equipped to mitigate this issue. We then analyze a simple teacher-student task (Section 4). This task already reveals the remarkable complexity underlying the learning of linear recurrent networks and enables us to verify empirically our theory. Finally, we discuss how our findings extend to more realistic scenarios (Section 5), both in terms of architectures and data. Overall, our paper provides theoretical insights into the training of recurrent neural networks, an area where such analysis is rare. While vanishing and exploding gradients are well-known challenges, our results demonstrate that this is not the end of the story - there exists an additional layer of complexity beyond them.

38th Conference on Neural Information Processing Systems (NeurIPS 2024).

# 1 Vanishing and exploding gradients

Let us first introduce the notations we will be using throughout the rest of the paper. We consider a recurrent neural network with hidden state $h_t$, update function $f_\theta$ parametrized by $\theta$, and input sequence $(x_t)$. The average performance of the network is measured by a loss $L$. We have

$$h_{t+1} = f_\theta(h_t, x_{t+1}) \text{ and } L = \mathbb{E}\left[\sum_{t=1}^{T} L_t(h_t)\right]. \tag{1}$$

The gradient of the instantaneous loss $L_t$ with respect to the parameters $\theta$ is then equal to

$$\frac{\mathrm{d}L_t}{\mathrm{d}\theta} = \frac{\partial L_t}{\partial h_t}\frac{\mathrm{d}h_t}{\mathrm{d}\theta} = \frac{\partial L_t}{\partial h_t}\sum_{t'\leq t}\frac{\mathrm{d}h_t}{\mathrm{d}h_{t'}}\frac{\partial f_\theta}{\partial\theta}(h_{t'-1}, x_{t'}) \tag{2}$$

In the equation above, we used $\partial$ to denote partial derivatives and $\mathrm{d}$ for total derivatives. Using this notation enables us to distinguish between $\partial_{h_t} L_t$, which corresponds to the error backpropagated from the current loss term to the hidden state through the readout function, and $\mathrm{d}_{h_t} L$, which accumulates the errors that are backpropagated through the future hidden state values. In particular, $\partial_{h_t} L = \partial_{h_t} L_t$ and $\mathrm{d}_{h_t} L = \partial_{h_t} L_t(h_t) + \sum_{t'>t} \mathrm{d}_{h_t} L_{t'}(h_{t'})$. When stacking several recurrent layers on top of each other, $\partial_{h_t} L$ corresponds to the current error being backpropagated to the hidden state $h_t$ through the hierarchy of the network and $\mathrm{d}_{h_t} L$ to future error signals backpropagated through the recurrence.

Early work [5] highlighted the difficulty for gradient descent to make recurrent neural networks remember past inputs that will later be useful to produce a desired behavior. This is due to the fact that error signals flowing backward in time tend to either explode or vanish. The key quantity is

$$\frac{\mathrm{d}h_t}{\mathrm{d}h_{t'}} = \prod_{i=t'}^{t-1}\frac{\partial h_{i+1}}{\partial h_i} = \prod_{i=t'}^{t-1}\frac{\partial f_\theta}{\partial h}(h_i, x_{i+1}). \tag{3}$$

One can remark that this quantity exponentially converges to 0 when the spectral radius of the Jacobian $\partial_h f_\theta$ is upper bounded by a constant strictly smaller than 1, and can exponentially explode if there exists some component bigger than 1. The error signal at time $t$ backpropagated to time $t'$ behaves similarly, as $\mathrm{d}_{h_{t'}} L_t = \partial_{h_t} L_t \, \mathrm{d}_{h_{t'}} h_t$. Gradient-based learning of long-term memories is thus difficult: the contribution of past hidden states to the current loss becomes either negligible or predominant as the time span considered increases.

Since then, the analysis has been refined [6–8] and the development of recurrent architectures has mostly been driven by the desire to solve this pathological issue. Most famously, the LSTM [3] unit, and later on the GRU [23], solve this problem by using memory neurons that facilitate direct information storage and retrieval, and thus facilitate error backpropagation. Other approaches to solving this problem, to name a few, involve gradient clipping [24, 8], activity normalization [25–27], careful weight initialization [28, 29] or enforcing architectural constraints such as hierarchical processing [30, 31], orthogonal weight matrices [32–34] and oscillations [35–37].

# 2 The curse of memory

According to common deep learning wisdom, it is often believed that solving the vanishing and exploding gradients problem enables recurrent neural networks to learn long-term dependencies. We challenge this view and question: is solving those issues really enough to ensure well-behaved loss landscapes? We answer negatively by showing that gradients can explode as the memories of the network are kept for longer, even when the dynamics of the network remains stable.

## 2.1 Intuition

Recurrent neural networks have something special: the very same update function $f_\theta$ is applied over and over. Therefore, modifying the parameters $\theta$ will not only influence one update, as changing the weights of a given layer in a feedforward neural network would, but all of them. As the memory of the network gets longer, the hidden states keep a trace of the effects of more updates. Hidden states thus become increasingly sensitive to parameter changes. This is the *curse of memory*. We borrow

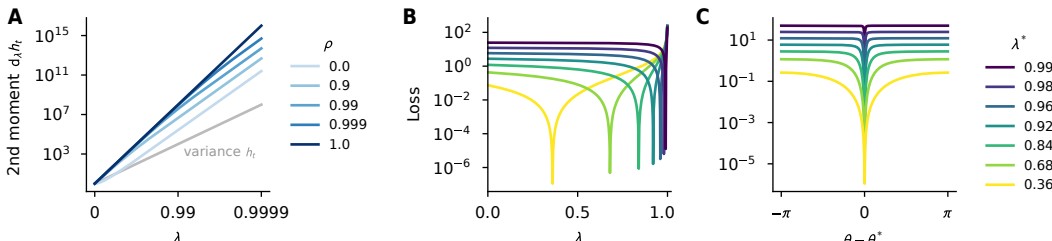

Figure 1: **Optimization of recurrent neural networks gets harder as their memory increases.**
**A.** Evolution of the second moment of $d_\lambda h_t$ as a function of the recurrent parameter $\lambda$ and of the input $x$ auto-correlation decay rate $\rho$, when $h_{t+1} = \lambda h_t + x_t$. As the memory of the network increases ($\lambda \to 1$), $h_t$ becomes more sensitive to changes in $\lambda$, particularly as the elements in the input sequence are more correlated ($\rho \to 1$). The explosion of $d_\lambda h_t$ is faster than the one of $h_t$, as highlighted with the grey line obtained for $\rho = 1$. See Section 2.2 for more detail. **B, C.** Illustration of the phenomenon on the toy one-dimensional teacher-student task of Section 4.1, in which the teacher is parametrized by a real number $\lambda^*$ and the student by a complex number $\lambda$. In B, $\lambda$ varies on the real axis, and it varies on the circle of radius $\lambda^*$ parametrized by $\theta$ in C. The loss becomes sharper as information is kept longer in memory, making gradient-based optimization nearly impossible.

the term from [38, 39], although we use it in a different context, and note that Martens and Sutskever [40] hypothesized that such a phenomenon could arise in RNNs and hinder their optimization.

Let us formalize our intuition by considering the sensitivity of the hidden state $h_t$ on the parameters $\theta$:

$$\frac{\mathrm{d}h_t}{\mathrm{d}\theta} = \sum_{t' \leq t} \frac{\mathrm{d}h_t}{\mathrm{d}h_{t'}} \frac{\partial f_\theta}{\partial \theta}(h_{t'-1}, x_{t'}). \tag{4}$$

When information stays in the network's memory for longer, the number of non-negligible Jacobian $d_{h_{t'}} h_t$ terms increases. As a result, the magnitude of this sensitivity increases when the network encodes longer-term dependencies, and learning $\theta$ becomes trickier. It is crucial to observe that this phenomenon arises even when exploding gradients are removed from the picture by constraining the eigenvalues of the recurrent Jacobian to be smaller than one and ensuring that the network dynamics remains stable. The rest of this section will be dedicated to studying this behavior quantitatively.

## 2.2 Signal propagation in linear diagonal recurrent neural networks

We study how hidden state and gradient magnitudes evolve as the network encodes longer-term dependencies. Ideally, these quantities do not vanish or explode, as it improves the conditioning of the loss landscape [41] and eases optimization [42, 43]. We operate under the following assumptions:

a) **Linear diagonal recurrent neural networks.** We restrict ourselves to update functions of the form $f_\theta(h_t, x_{t+1}) = \lambda \odot h_t + x_{t+1}$ with $\lambda$ a vector of the size of $h_t$ and $\odot$ the element-wise product. For ease of exposition, we present results for real-valued $\lambda$ here; see Appendix B.2 for the complex-valued setting. While this assumption is strong, it allows us to identify some crucial mechanisms and models like S4 [17], S5 [19] and LRUs [20] satisfy it. We later show that our analysis can model some features of more sophisticated networks. Note that we do not consider the input and readout mappings usually featured in recurrent layers as they are feedforward layers and signal propagation within them is already well understood [e.g., 44, 45].

b) **Infinite time horizon.** We consider infinite sequences and initialize the network dynamics at $t_0 = -\infty$. It simplifies our calculations while being a reasonable assumption when the sequences considered are longer than the characteristic timescales of the dependencies we want to learn.

c) **Wide-sense stationarity.** We assume the different quantities that the network receives, which include the inputs $x_t$, to be wide-sense stationary (WSS). A random process $X_t$ is said to be WSS if its auto-correlation function is independent of time, that is, for all $t \in \mathbb{Z}$ and $\Delta \in \mathbb{Z}$, $\mathbb{E}_X[X_{t+\Delta}X_t] =: R_X(\Delta)$, where $\mathbb{E}_X$ denotes the expectation over the data. It corresponds to assuming that the statistics of the data are invariant to time shifts. This is a standard assumption when analyzing stochastic processes [46]. It keeps our calculations concise and does not qualitatively affect our conclusions (cf. Section 5). We discuss how to relax it in Appendix B.2.4.

We are now equipped to analyze signal propagation in one recurrent layer, both in the forward and backward passes. We show that both hidden states and backpropagated errors explode as $|\lambda| \to 1$.

**Forward pass.** Here, we are interested in understanding how the hidden state second moment $\mathbb{E}[h_t^2]$ evolves as a function of $\lambda$ and of the input auto-correlation function $R_x$. After a calculation that we defer to Appendix B.2, we obtain

$$\mathbb{E}\left[h_t^2\right] = \frac{1}{1-\lambda^2}\left(R_x(0) + 2\sum_{\Delta \geq 1}\lambda^\Delta R_x(\Delta)\right). \tag{5}$$

Importantly, this quantity goes to infinity as longer-term dependencies are encoded within the network, that is $|\lambda| \to 1$. Additionally, the divergence speed depends on the input data distribution: it increases as consecutive time steps in the input distribution become more correlated (i.e., less of the $R_x(\Delta)$ terms are negligible). This behavior already highlights potential difficulties of gradient-based learning of deep neural networks containing linear recurrent layers as the variance of neural activity can become arbitrarily large, hindering learning abilities of deeper layers.

**Backward pass.** Let us first derive the gradient of the loss with respect to $\lambda$. Using the chain rule we have $\mathrm{d}_\lambda L = \sum_t \partial_{h_t} L \, \mathrm{d}_\lambda h_t$. We thus seek to understand how $\mathrm{d}_\lambda h_t$ behaves. We remark that $\mathrm{d}_\lambda h_{t+1} = \lambda \mathrm{d}_\lambda h_t + h_t$ so that $\mathrm{d}_\lambda h_t$ is a low pass filtered version of the hidden state, which is itself a low pass filter version of the inputs. It therefore comes as no surprise that the second moment of $\mathrm{d}_\lambda h_t$ diverges faster than the one of $h_t$ when $|\lambda| \to 1$. More precisely, we get

$$\mathbb{E}\left[\left(\frac{\mathrm{d}h_t}{\mathrm{d}\lambda}\right)^2\right] = \frac{1+\lambda^2}{(1-\lambda^2)^3}\left(R_x(0) + 2\sum_{\Delta \geq 1}\lambda^\Delta R_x(\Delta)\right) + \frac{2}{(1-\lambda^2)^2}\left(\sum_{\Delta \geq 1}\Delta\lambda^\Delta R_x(\Delta)\right). \tag{6}$$

We plot the exact behavior of this quantity when the auto-correlation of $x$ satisfies $R_x(\Delta) = \rho^{|\Delta|}$ on Figure 1 and refer the interested reader to Appendix B.2 for a derivation of Equation 6. More generally, the hidden state of the network, and thus its final output, becomes increasingly sensitive to changes in recurrent parameters as the network reaches the edge of dynamical stability ($|\lambda| \to 1$).

The last quantity we need to consider is the error that is backpropagated to the inputs $x$ of the recurrent layer. It can be observed that the backward pass is dual to the forward pass in the sense that it is a recurrent process that receives backpropagated errors $\partial_{h_t} L$ and it runs in reverse time:

$$\frac{\mathrm{d}L}{\mathrm{d}x_t} = \frac{\mathrm{d}L}{\mathrm{d}h_t}\frac{\partial h_t}{\partial x_t} = \frac{\mathrm{d}L}{\mathrm{d}h_{t+1}}\frac{\partial h_{t+1}}{\partial h_t} + \frac{\partial L}{\partial h_t} = \lambda\frac{\mathrm{d}L}{\mathrm{d}h_{t+1}} + \frac{\partial L}{\partial h_t}, \tag{7}$$

in which we made use of $\partial_{x_t} h_t = 1$. It follows that the analysis we did for the forward pass also holds here. Crucially, this implies that the explosion behavior will be most significant for the recurrent parameters rather than for potential input or readout weights.

### 2.3 Extending the analysis to the non diagonal case

We now generalize our results to fully connected linear recurrent neural networks of the form $h_{t+1} = Ah_t + x_t$. For the sake of the analysis, we assume that $A$ is complex diagonalizable, that is there exists a complex-valued matrix $P$ and a complex-valued vector $\lambda$ such that $A = P\mathrm{diag}(\lambda)P^{-1}$. Note that this occurs with probability one under random initialization of $A$ [20]. In this case,

$$h_t = Ph_t^{\mathrm{diag}} \quad \text{with } h_{t+1}^{\mathrm{diag}} = \mathrm{diag}(\lambda)h_t^{\mathrm{diag}} + P^{-1}x_{t+1} \tag{8}$$

and

$$\frac{\mathrm{d}h_t}{\mathrm{d}A} = \frac{\partial h_t}{\partial P}\frac{\partial P}{\partial A} + \frac{\partial h_t}{\partial h_t^{\mathrm{diag}}}\frac{\mathrm{d}h_t^{\mathrm{diag}}}{\mathrm{d}\lambda}\frac{\partial\lambda}{\partial A} + \frac{\partial h_t}{\partial h_t^{\mathrm{diag}}}\frac{\mathrm{d}h_t^{\mathrm{diag}}}{\mathrm{d}P^{-1}}\frac{\partial P^{-1}}{\partial A}. \tag{9}$$

From the analysis above, we know that the dominating term in the limit $|\lambda| \to 1$ among $\partial_P h_t$, $\mathrm{d}_\lambda h_t$ and $\mathrm{d}_P^{-1} h_t$ is $\mathrm{d}_\lambda h_t$, as $P$ and $P^{-1}$ act as readout and input weights. Given that all other terms do not directly depend on the magnitude of $\lambda$, we have that $\mathrm{d}_A h_t \simeq \partial_{h_t^{\mathrm{diag}}} h_t \, \mathrm{d}_\lambda h_t^{\mathrm{diag}} \partial_A \lambda$; cf. Appendix B.2.3 for formal statements. This has two consequences: First, the sensitivity of $h_t$ on $A$ will explode

as longer memories are encoded and this directly comes from the eigenvalues of $A$. Second, as each entry of $A$ typically impacts all eigenvalues of the matrix, the explosion behavior will be distributed across all entries, whereas it was concentrated on the eigenvalues for the diagonal case. We will later observe that this has significant practical consequences and partly explains why fully connected linear RNNs are difficult to train. As a side note, we remark that enforcing the matrix $A$ to be orthogonal solves vanishing and exploding gradient issues but these weights may remain sensitive to learn because of the curse of memory.

## 3  Mitigating the curse of memory

We have discussed the sensitivity of recurrent networks to parameter updates. Given this problem, how can it be mitigated? We show that recurrent networks with diagonal connectivity are particularly well suited for this purpose. Besides enabling control over the Jacobian and avoiding exploding gradients, they facilitate the mitigation of the curse of memory. We additionally highlight that deep state-space models and gated RNNs inherently incorporate such mechanisms.

### 3.1  A solution: normalization and reparametrization

Both forward and backward passes explode as the network encodes longer memories. When $h_{t+1} = \lambda h_t + x_{t+1}$, we argue that it is relatively straightforward to mitigate this effect. We aim to keep $\mathbb{E}[h_t^2]$, $\mathbb{E}[(\mathrm{d}_\lambda h_t)^2]$ and $\mathbb{E}[(\mathrm{d}_{x_t} h_t)^2]$ independent of $\lambda$, similar to initialization schemes that maintain the magnitude of neural activity constant in deep networks [44, 45], regardless of the layer width [42, 47, 43].

**Input normalization.** A simple way to enforce $\mathbb{E}[h_t^2]$ to stay constant is to introduce a scaling factor $\gamma(\lambda)$ applied to the inputs a neuron receives, that satisfies $\gamma(\lambda)^2\mathbb{E}[h_t^2] = \Theta(1)$. Given that the backward propagation of output errors to inputs is dual to the forward pass, the role of $\gamma$ in the backward pass will be similar. The value $\gamma$ needs to take therefore both depends on the input distribution to normalize the forward pass, as well as on the output error distribution to normalize the backward pass. Perfect normalization is likely unrealistic, but some normalization can help, as shown in Figure 2.A.

**Eigenvalue reparametrization.** We are now left with keeping the gradient of the loss with respect to $\lambda$ under control. Input normalization partly reduces the memory-induced exploding effect, but not entirely as the variance of $\mathrm{d}_\lambda h_t$ is much larger than the one of $h_t$ (cf. Fig.1.A). Reparametrization can close that gap. Indeed, if $\lambda$ is parametrized by $\omega$, we have that $\mathrm{d}_\omega h_t = \mathrm{d}_\lambda h_t \mathrm{d}_\omega \lambda$. Choosing a parameterization that is more and more granular as $\lambda$ goes to 1 thus helps in keeping the magnitude of $\mathrm{d}_\omega h_t$ constant. Assuming $\gamma$ is independent of $\lambda$ for simplicity, achieving $\mathbb{E}[(\mathrm{d}_\omega h_t)^2] = \Theta(1)$ requires solving the differential equation $\gamma(\lambda)^2\lambda'(\omega)^2\mathbb{E}[(\mathrm{d}_\lambda h_t)^2] = 1$. While deriving a universal optimal parametrization is again unrealistic due to dependency on the input distribution, reparametrization definitely helps, as shown in Figure 2.B. Figure 6 illustrates how it affects the loss landscape.

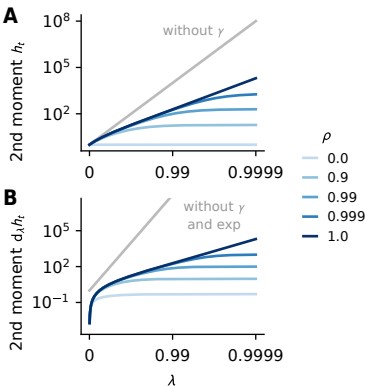

Figure 2: **Illustration of the effects of normalization and reparametrization.** It can effectively control the magnitude of **A.** $\mathbb{E}[h_t^2]$ and **B.** $\mathbb{E}[(\mathrm{d}_\lambda h_t)^2]$ over all $\lambda$ values when the input auto-correlation satisfies $R_x(\Delta) = \rho^{|\Delta|}$ with $\rho = 0$, but does not manage do to so for other type of distributions ($\rho \neq 0$). Here, we use $\gamma(\lambda) = \sqrt{1 - \lambda^2}$, decouple it from $\lambda$ when differentiating, and take $\lambda = \exp(-\exp(\nu))$, as in [20]. The grey line indicates the value the two quantities take without any normalization and reparametrization, when $\rho = 1$.

**The case of complex numbers.** We have not yet discussed the case $\lambda \in \mathbb{C}$, relevant for SSMs such as S4 [17]. We extend our analysis to complex-valued $\lambda$ in Appendix B.3.2. Briefly, it reveals that changes in the magnitude of $\lambda$ have a similar impact as in the real case, but this similarity does not extend to the angle. To keep the sensitivity on the angle constant, its parametrization must depend on the magnitude of $\lambda$. However, doing so hurts learning, particularly far from optimality, as we exemplify in Appendix B.3.2. A key implication of this analysis is that the sensitivity of the hidden state on the angle of $\lambda$ explodes as its magnitude approaches 1.

## 3.2 Several RNN architectures implicitly alleviate the curse of memory

Deep state-space models, as well as gated RNNs, feature some form of normalization and reparametrization which help keeping signal propagation under control. We discuss how below.

**Deep state-space models.** Deep SSMs were originally motivated as discretizations of the differential equation $\dot{h} = Ah + Bx$ [16]. Naïve discretization of the differential equation yields $h_{t+1} = (\text{Id} + \text{d}tA)h_t + \text{d}tBx_{t+1}$ which already provides some input normalization. More elaborate discretization schemes, such as the zero-order hold, effectively reparametrize the $A$ matrix, e.g. with $\exp(\text{d}tA)$. Here, diagonalization arises from computational efficiency and simplicity reasons [18]. While such models can approximate any smooth mappings [48, 49], their expressivity remains limited [50]. The next generation of these models, including Mamba [21], incorporates input-dependent gates which modulate $\text{d}t$ depending on the input $x_t$. The theory we developed above does not strictly apply to this setting as $\text{d}t$ is no longer constant. However, since the recurrence Jacobian remains diagonal, we expect the qualitative behaviors we analyzed to remain.

**Gated RNNs.** While the original motivation behind gated RNNs such as LSTMs [3] or GRUs [23] largely differs from the one of SSMs, they share similar mechanisms. In these networks, the memories stored in hidden neurons can be erased through a forget gate, and incoming inputs can selectively be written in memory through an input gate. Mathematically, this corresponds to hidden state updates of the form $h_{t+1} = f_{t+1} \odot h_t + i_{t+1} \odot x_{t+1}$, with the forget $f_{t+1}$ and input $i_{t+1}$ gates being independent non-linear functions of $x_{t+1}$ and $h_t$. The forget gate is akin to $\lambda$ and usually involves a sigmoid non-linearity, which has a similar effect as reparametrizing $\lambda$ in the backward pass. The input gate can act as an input normalization depending on the initialization of the network or if is coupled to the forget gate as in the GRU ($f_t = 1 - i_t$) [29]. Importantly, the gates here depend on the hidden states and thus make the Jacobian $\partial_{h_t} h_{t+1}$ non diagonal. Yet, we argue that these architectures still have a bias towards diagonality. Indeed, the contributions of the hidden state through the forget and input gates are indirect, and they can be ignored when the weights connecting the hidden states to the gates are small. Empirically, we find that GRUs lie in this regime at initialization, cf. Section D.2, so that our theory accurately captures signal propagation in GRUs. We additionally confirm that signal propagation is well behaved in gated RNNs in Section 5. In regimes in which this approximation does not hold, studying signal propagation requires a much more sophisticated analysis than the one we have done here [51].

## 4 A linear teacher-student analysis

We start our empirical analysis with a teacher-student task using linear recurrent networks [52]. This is arguably the simplest setting in which one can train recurrent networks. Yet, as we shall see, it is already remarkably complex and captures some of the differences between different architectures observed in more realistic settings [20]. It additionally makes it possible to control the characteristic time constants of the data, which is only possible with synthetic data.

Our investigation starts with the one-dimensional setting to provide an intuitive illustration of the consequences of the curse of memory on the loss landscape. We then address the general setting and observe that linear networks indeed suffer from the curse of memory, and that the remedies we studied in the last section are effective. We additionally find that diagonality greatly modifies the structure of the loss landscape and helps optimizers with adaptive learning rates to compensate for eventual increased sensitivities.

### 4.1 The one-dimensional case

We first consider a student and a teacher following the one-dimensional dynamics $h_{t+1} = \lambda h_t + x_{t+1}$, with complex-valued parameter $\lambda$ for the student and $\lambda^*$ for the teacher. For simplicity, we independently draw each $x_{t+1}$ from a unit normal distribution (its autocorrelation function is $R_x(\Delta) = \delta_{\Delta=0}$) and note that other input distributions do not qualitatively change the results. The performance of the student is measured by a loss $L$ that averages the per time-step losses $L_t := \frac{1}{2}|h_t - h_t^*|^2$ over the entire sequence.

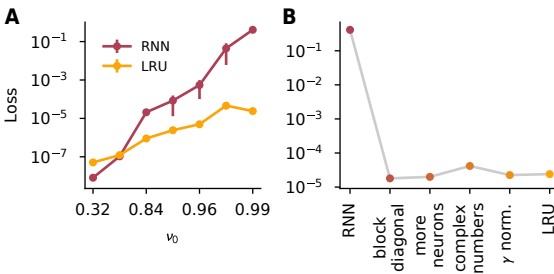

Figure 3: **LRUs are better at replicating a teacher's behavior than linear RNNs. A.** As the teacher encodes longer dependencies ($\nu_0 \to 1$), the linear RNN struggles to reproduce it, but not the LRU. **B.** An ablation study ($\nu_0 = 0.99$) reveals that this gap mainly comes from having a close to diagonal recurrent connectivity matrix. See Section 4.2 for more detail.

This simple model already captures two key difficulties of gradient-based learning of recurrent neural networks. In Figure 1, we plot the resulting loss landscape for different $\lambda^*$ values, when $\lambda$ evolves on the positive part of the real axis (Fig. 1.B) and when it evolves on the circle of radius $|\lambda^*|$ in the complex plane (Fig. 1.C). We restrict $\lambda$s to have absolute values smaller than one: exploding gradients are out of the picture. Still, two difficulties for gradient-based learning appear here. On one side, vanishing gradients lead to flat loss regions that are hard to escape. On the other side, the loss sharpens as the student encodes longer memories because of the curse of memory. As a consequence, gradient-based optimization is extremely tedious, already in this simple example.

## 4.2 Diagonal connectivity simplifies optimization

We now move to the general case in which the teacher evolves according to

$$h_{t+1} = Ah_t + Bx_{t+1} \ \text{ and } \ y_t = Ch_t + Dx_t. \tag{10}$$

with $h_t \in \mathbb{R}^n$, $x_t \in \mathbb{R}$ drawn i.i.d. from $\mathcal{N}(0,1)$, $A \in \mathbb{R}^{n \times n}$, $B \in \mathbb{R}^{n \times 1}$, $C \in \mathbb{R}^{1 \times n}$ and $D \in \mathbb{R}^{1 \times 1}$. Here both inputs and outputs are scalars.

Given the intuition we have developed so far, we expect fully connected linear recurrent neural networks to struggle solving the task when the teacher encodes longer memories, not only because of exploding gradients but also due to the curse of memory. Conversely, diagonality facilitates the eigenvalue reparametrization needed to avoid exploding gradients and make them better behaved. We run the following experiment to verify this intuition. We draw random teachers with hidden dimension $n = 10$ and transform the complex eigenvalues of the recurrent matrix $A$ to have magnitudes close to a value $\nu_0$ that we control[1]. The larger $\nu_0$ is, the longer the memories encoded by the teacher are. We train a linear RNN, as well as an LRU [20], with hidden dimension $64$ on this task. The students are therefore largely overparametrized. We chose the LRU architecture to represent deep SSMs due to its simplicity. This architecture uses input normalization and an exponential reparametrization of the eigenvalues, similar to what we analyze in Section 3. Both networks are trained using the Adam optimizer [53] and cosine annealing schedule for 10k steps, on batches of size $128$. To ensure that we are in the infinite sequence length regime, we take the sequences to be of length $300$, that is three times longer than the characteristic time scale of the teacher. Learning rates are tuned separately for each method and training distribution. The results, which we plot in Figure 3.A, confirm our intuition: LRUs significantly outperform linear RNNs when long memories have to be learned, despite having 10 times fewer parameters.

Next, we wonder which design choices behind the LRU architecture are crucial to this performance improvement. To this end, we interpolate between a linear RNN and an LRU in the following way: First, we restrict the weight matrix of the linear RNN to a block diagonal with blocks of size 2. Each block can represent a complex number, so the network can represent 32 complex numbers in total. We additionally double the number of hidden neurons. Second, we change those $2 \times 2$ blocks (and their input and output weights) to be complex numbers. Finally, we add the $\gamma$ input normalization and the exponential parametrization to obtain the final LRU architecture. We report the results of this experiment in Figure 3.B. Surprisingly, we find the gap comes from making the weight matrix block diagonal ($4 \times 4$ blocks are here enough, cf. Figure 11 in the Appendix). Interestingly, this change reduces the number of parameters the model has and slightly reduces the model expressivity. An explanation of this behavior is therefore likely to be related to the optimization properties of those models. We confirm this hypothesis in the next section.

---

[1]We draw each entry of $A$ from $\mathcal{N}(0, 1/\sqrt{n})$, complex diagonalize it, and apply the transformation $x \mapsto \nu_0 + (1 - \nu_0)\tanh(x)$ to the absolute values of the eigenvalues.

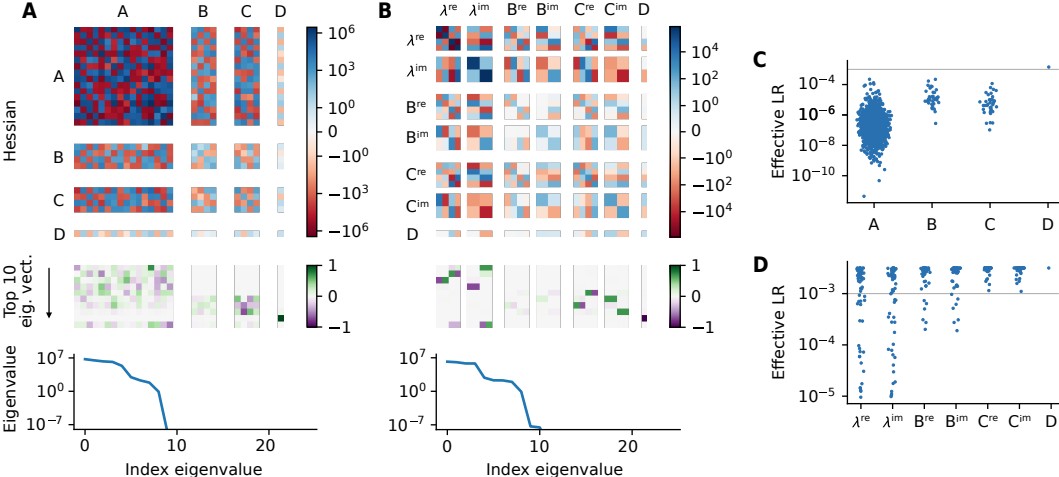

Figure 4: **Differences in learning abilities between fully connected and complex diagonal linear RNNs are due to a better structure of the loss landscape. A, B.** Hessian of the loss at optimality, its 10 eigenvectors with greatest eigenvalues and its eigenspectra for a fully connected RNN (A) and a complex diagonal one (B). The spectra are almost the same. However, the top eigenvectors are concentrated on few coordinates for the complex diagonal one but not for the fully connected one. **C, D.** This structure makes it possible for Adam to efficiently deal with the extra sensitivity, as shown with the effective learning rates that it uses at the end of learning. For the fully connected one (C), Adam uses small learning rates to compensate for the sensitivity, whereas it can use larger ones for the complex diagonal one without hindering training stability. The horizontal grey line shows the learning rate used, which is here $10^{-3}$.

## 4.3 On the importance of adaptive learning rates

So far, our results highlight the importance of having a close to diagonal recurrent connectivity matrix. In this section, we show that this parametrization alone does not mitigate any exploding behavior but modifies the structure of the loss landscape, making it possible for optimizers with adaptive learning rates to compensate for these behaviors.

To demonstrate this, we consider the Hessian of the loss:

$$\frac{\mathrm{d}^2 L}{\mathrm{d}\theta^2} = \sum_t \mathbb{E}_x \left[ \frac{\mathrm{d}h_t}{\mathrm{d}\theta} \frac{\partial^2 L_t}{\partial h_t^2} \frac{\mathrm{d}h_t}{\mathrm{d}\theta}^\top + \frac{\partial L_t}{\partial h_t} \frac{\mathrm{d}^2 h_t}{\mathrm{d}\theta^2} \right]. \tag{11}$$

If the network can perfectly fit the target data, which is the case in the experiments above, the second term vanishes at optimality. We plot the Hessian at optimality in Figure 4.A and B for a standard linear recurrent network and one with complex diagonal parametrization, both with 4 hidden neurons ($\nu_0 = 0.99$). We observe that the eigenvalue spectra are similar for the two architectures, both exhibiting large terms that are characteristic of the curse of memory, which makes learning with stochastic gradient descent almost impossible[2]. However, their structures differ. For the fully connected linear RNN, the top eigenvectors are distributed over many coordinates, whereas they are concentrated on a few coordinates for the complex diagonal one. This feature aids adaptive optimization [e.g., 57]: adapting to large curvature is much easier for Adam when the pathological directions are aligned to the canonical basis. This is what we observe in practice.

In Figure 4.C and D, we compare the effective learning rate used by Adam, which we compute by providing a vector of ones to the optimizer. For the dense linear RNN, the adaptive learning rates cannot compensate for the intricate coupling between components, resulting in small learning rates overall. Conversely, the sensitive directions of complex diagonal RNNs are concentrated on few

---

[2]The gradient Lipschitz constant $L$ of the loss equals the maximum Hessian eigenvalue [54]. This quantity sets a bound $2/L$ for the maximum globally stable learning rate. While convergence might happen in a subspace, it is generally aligned with the top Hessian eigenspace near the solution [55, 56].

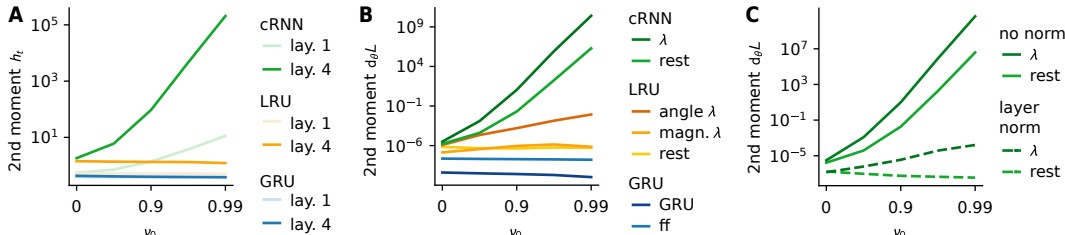

Figure 5: **Signal propagation in deep recurrent networks at initialization is consistent with our theory.** **A.** $\mathbb{E}[h_t^2]$ after the first and the fourth layer, as a function of the exponential decay parameter $\nu_0$, for complex-valued diagonal RNN (cRNN), LRU, and GRU recurrent layers. The input normalization present in the LRU and in the GRU effectively keeps neural activity constant across $\nu_0$ values. **B.** Comparison of the evolution of the loss gradient $\mathbb{E}[(d_\theta L)^2]$ for the different recurrent layers and specific groups of parameters. For the complex diagonal RNN, the gradients of all parameters explode and in particular the ones of the recurrent parameters, whereas only the ones of the angle of $\lambda$ explode for the LRU, consistently with the theory. Error signal propagation in GRUs is under control: the magnitude of the gradients is independent of $\nu_0$. The GRU-specific parameters exhibit smaller gradients than the feedforward (ff) ones. **C.** Layer normalization keeps the overall gradient magnitude under control in cRNNs. Batch normalization yields similar results.

parameters, which adaptive learning rates can compensate for. This leads to more targeted and overall larger learning rates, significantly speeding up learning. As a side note, the complex eigenvalues of the teacher come in conjugate pairs. However, during training, the complex values of the complex RNN are not conjugates of each other, thereby increasing Hessian diagonality. Finally, performing this analysis for the LRU, we find that the Hessian spectrum is similar to the diagonal setting and that the exploding dimensions of the Hessian are almost exclusively due to the angle parameter, consistently with our theoretical analysis; see Figure 9.A and C. The loss landscape for S4 model [17] can be qualitatively similar to the complex diagonal RNN or to the LRU, depending on which regime it is in; see Figure 9.B and D.

Before concluding this section, we investigate whether certain eigenvalue distributions can break the diagonal structure of the Hessian, thereby complicating optimization and increasing the need for eigenvalue reparametrization. In Appendix C.2, we provide a theoretical quantification of the intuitive result that more concentrated eigenvalues lead to less diagonal Hessian. Consequently, the performance gap between complex-valued diagonal networks and LRUs widens, although the former still greatly outperform their fully-connected counterpart (see Figure 10). An important corollary is that increasing the number of hidden neurons breaks the diagonal structure of the loss landscape, thus reducing the effectiveness of optimizers with adaptive learning rates in mitigating the curse of memory. This observation may explain why Orvieto et al. [20] reported a more substantial performance improvement from eigenvalue reparametrization than what we observe in our study (cf. block diagonal vs. LRU in Figure 3.B).

## 5   Signal propagation in deep recurrent networks at initialization

The ultimate goal of our theoretical quest is to gain insights into the training of practical recurrent network architectures. Specifically, we aim to verify whether the trends established theoretically and in controlled experiments hold in practice, by studying signal propagation at initialization on a realistic next-token prediction natural language processing task.

To that matter, we provide sentences as input to deep recurrent networks that contain four blocks and use a next-token prediction loss to measure their performance. Each block consists of a recurrent layer followed by a feedforward gated linear unit [58]. By default, there are no normalization layers in this architecture. More details can be found in Appendix D.1. We empirically study how $\mathbb{E}[h_t^2]$ and $\mathbb{E}[(d_\theta L)^2]$ evolve when the characteristic time scale of the recurrent layers, controlled through $\nu_0$, increases. We compare three different recurrent layers: a complex-valued diagonal RNN (cRNN), a LRU and a GRU initialized with the Chrono initialization [29].

The results are consistent with our theory. Complex-valued RNNs suffer from the curse of memory and recurrent parameters grow faster to infinity than the rest as $\nu_0$ goes to 1. Perhaps more surprisingly,

a finer grain analysis reveals that the gradient magnitude is independent of the layer; see Figure 13. LRUs almost perfectly mitigate exploding behaviors in the forward pass (Figure 5.A) as well as in the backward pass (Figure 5.B), except for the angle parameter, consistently with our previous analysis. We also wonder whether layer normalization can replace the input normalization and reparametrization of the LRU. We find that it mitigates the memory-induced gradient explosion at the macroscopic level (Figure 5.C), but it likely kills any learning signal for the smallest eigenvalues [20]. Finally, the GRU manages to keep the gradient magnitude constant over different characteristic time constants, consistently with the intuition we developed in Section 3.2. Preliminary experiments revealed that same trends also hold for LSTMs.

The results presented above for the GRU align qualitatively with the intuition developed throughout the paper. We now consider how well our theory can quantitatively explain this behavior. The primary difference between our simple model and GRUs is that the $\lambda$ values, referred to as forget gates in the GRU terminology, depend on both inputs and hidden states, and are therefore not constant. Interestingly, we find that GRUs almost behave like the diagonal linear RNNs we have focused on in this paper, particularly for slowly decaying recurrent neurons with high $\nu_0$ values (see Appendix D.2). Consequently, applying our theory as if this context-dependency does not exist only introduces minor approximation errors, which we confirm empirically in Appendix D.3. Given the similarity of the Chrono initialization to those used in modern architectures like Mamba [21] and Hawk [22], we expect our theory to also serve as a good proxy for studying signal propagation in these models at initialization.

# 6   Conclusion

Vanishing and exploding gradients complicate the learning of recurrent networks, but solving these problems is not enough. We uncovered yet another difficulty of training such networks, which is rooted in their iterative nature and arises at the edge of dynamical stability. Reparametrizations and adaptive learning rates can effectively mitigate this behavior in practice, and diagonalizing the recurrence simplifies both. Our analysis additionally reveals the complexity of learning the angle of complex eigenvalues, which may explain why complex numbers were not found to be useful in most recent state-space model architectures [21, 22].

A side finding of our study is the symbiosis between independent modules, which are here neurons and can be more generally small heads, with adaptive learning rate optimizers in linear recurrent networks. Such a design pattern has promising properties: it facilitates online learning [59] and compositional generalization [60], allows for high level of parallelization [22], and matches, at a high level, the modular organization of the cortex in cortical columns [61]. Understanding how to increase the expressivity of small linear modules while keeping their great optimization properties constitutes a promising avenue for future research.

## Limitations

The theory we introduced, with its focus on signal propagation, only addresses the training dynamics of recurrent neural networks. Consequently, it does not provide insights into other important questions such as generalization abilities or memory capacities of these networks. The main assumption underlying this analysis is that the recurrence is both diagonal and linear. While this approach offers valuable insights, it can only approximate signal propagation in more sophisticated architectures. Given the simplicity of the analytical tools employed, there is little hope that this framework can be extended to more general settings without significant modifications.

## Acknowledgments

The authors thank Robert Meier, João Sacramento, Guillaume Lajoie, Ezekiel Williams, Razvan Pascanu, Imanol Schlag and Bobby He for insightful discussions. Nicolas Zucchet was supported by an ETH Research Grant (ETH-23 21-1) and Antonio Orvieto acknowledges the financial support of the Hector Foundation.

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

# Appendix

## Table of contents

# A    Definition of the recurrent networks we use

In this section, we rigorously define all the architectures we use in the main text and in the appendix, as well as precisely describe how we initialize them.

## A.1    Linear recurrent neural network

Let us start by introducing the linear recurrent neural network we are using. It satisfies

$$h_0 = 0 \tag{12}$$
$$h_{t+1} = A h_t + B x_{t+1} \tag{13}$$
$$y_t = C h_t + D x_t \tag{14}$$

with $x_t \in \mathbb{R}^{d_{\text{in}}}$, $h_t \in \mathbb{R}^n$, $y_t \in \mathbb{R}^{d_{\text{out}}}$, $A \in \mathbb{R}^{n \times n}$, $B \in \mathbb{R}^{n \times d_{\text{in}}}$, $C \in \mathbb{R}^{d_{\text{out}} \times n}$ and $D \in \mathbb{R}^{d_{\text{out}} \times d_{\text{in}}}$. We draw the entries of $B$, $C$ and $D$ from independent truncated normal distributions with fan_in scaling. We draw each entry of $A$ from $\mathcal{N}(0, 1/\sqrt{n})$ independently and then apply the following postprocessing to it: First we complex diagonalize $A$, which we can do almost surely. Note $\lambda$ its eigenvalues. We then transform them according to

$$\lambda \leftarrow (\nu_0 + (1 - \nu_0) \tanh(|\lambda|)) \exp\left(i \frac{\text{angle}(\lambda)}{\pi} \theta_0\right) \tag{15}$$

with $\nu_0$ and $\theta_0$ two scalars that we control. This transformation has several benefits: we are guaranteed that the magnitude of $\lambda$ is within $[\nu_0, 1]$ (and in $[\nu_0, \nu_0 + (1 - \nu_0) \tanh(1)]$ in the limit $n \to \infty$ as the eigenvalues of $A$ stay within the unit circle in that limit), and conjugate pairs of eigenvalues remain conjugate. This last point ensures that the resulting matrix remains real without having to change the eigenvectors. When we split the connectivity matrix $A$ into several independent blocks, we initialize each block separately with the scheme described above.

## A.2    Complex-valued RNN and linear recurrent unit (LRU)

Both architectures have recurrence of the form

$$h_0 = 0 \tag{16}$$
$$h_{t+1} = \lambda \odot h_t + \gamma \odot B x_t \tag{17}$$
$$y_t = \text{Re}[C h_t + D x_t] \tag{18}$$

with $\odot$ the element-wise product, $x_t \in \mathbb{R}^{d_{\text{in}}}$, $h_t \in \mathbb{C}^n$, $y_t \in \mathbb{R}^{d_{\text{out}}}$, $\lambda \in \mathbb{C}^n$, $B \in \mathbb{C}^{n \times d_{\text{in}}}$, $\gamma \in \mathbb{R}^n$, $C \in \mathbb{C}^{d_{\text{out}} \times n}$ and $D \in \mathbb{R}^{d_{\text{out}} \times d_{\text{in}}}$.

For the complex-valued linear RNN, we take

$$\lambda = \lambda^{\text{re}} + i \lambda^{\text{im}} \tag{19}$$
$$\gamma = 1 \tag{20}$$

so that it can be considered as parametrizing the diagonalized version of the linear RNN.

For the LRU [20], we take

$$\lambda = \exp(- \exp(\omega_\nu)) \exp(i \exp(\omega_\theta)) \tag{21}$$
$$\gamma = \exp(\omega_\gamma). \tag{22}$$

For both architectures, we use the LRU initialization so that $\lambda$ is uniformly distributed on the ring between the circles of radii $\nu_0$ and 1, with absolute angle restricted to be below $\theta_0$. For the LRU, we initialize $\gamma$ to $\sqrt{1 - |\lambda|^2}$.

## A.3    S4

We consider a slightly modified version of S4 here:

$$\Delta = \text{softplus}(\omega_\Delta) \tag{23}$$
$$h_{t+1} = \exp\left((\omega_A^{\text{re}} + i \omega_A^{\text{im}}) \Delta\right) \odot h_t + \Delta \odot B x_{t+1} \tag{24}$$
$$y_t = \text{Re}[C h_t + D x_t] \tag{25}$$

with $x_t \in \mathbb{R}^{d_{\text{in}}}$, $h_t \in \mathbb{C}^n$, $y_t \in \mathbb{R}^{d_{\text{out}}}$, $\omega_A^{\text{re}} + i\omega_A^{\text{im}} \in \mathbb{C}^n$, $B \in \mathbb{C}^{n \times d_{\text{in}}}$, $\omega_\Delta \in \mathbb{R}^n$, $C \in \mathbb{C}^{d_{\text{out}} \times n}$ and $D \in \mathbb{R}^{d_{\text{out}} \times d_{\text{in}}}$. The main difference with the standard architecture is that we do not consider any state expansion so that there is no parameter sharing. The makes this architecture closer to the linear RNN architecture we mostly focus on in this paper, while keeping the kind of parametrization it uses. In the same spirit, we do not couple the input and the readout matrices in any way.

The initialization we use also differs from existing ones as we initialize $\Delta$ to 1 and $\exp\left(\Delta(\omega_A^{\text{re}} + i\omega_A^{\text{im}})\right)$ in the same way as $\lambda$ in the LRU.

## A.4 GRU

The GRU version we use is the following:

$$r_{t+1} = \sigma(W_{rx}x_{t+1} + W_{rh}h_t + b_r) \tag{26}$$
$$f_{t+1} = \sigma(W_{fx}x + W_{fh}h_t + b_f) \tag{27}$$
$$n_{t+1} = \tanh(W_{nx}x_{t+1} + b_{nx} + r \odot (W_{nh}h_t + b_{nh})) \tag{28}$$
$$h_{t+1} = f_{t+1} \odot h_t + (1 - f_{t+1}) \odot n_{t+1} \tag{29}$$
$$\tag{30}$$

with $\sigma$ the sigmoid function, $h_t \in \mathbb{R}^n$, $x_t \in \mathbb{R}^{d_{\text{in}}}$ and all the other matrices appropriately sized. We initialize the parameters with the Chrono initialization [29]: all parameters are initialized using standard practice (orthogonal initialization for the weights taking $h$ as input, Lecun initialization for the rest, biases initialized at 0) except for $b_f$, which is initialized as

$$b_f \sim \log\left(\mathcal{U}\left(T_{\min}, T_{\max}\right)\right). \tag{31}$$

Here, $T_{\min}$ and $T_{\max}$ denote the minimum and maximal characteristic time scale. In the experiments of Section 5 we take

$$T_{\min} = \frac{1}{1 - \nu_0} \tag{32}$$

$$T_{\max} = \frac{2}{1 - \nu_0} \tag{33}$$

to enable the comparison with other architectures. Indeed, when ignoring the dependence of the forget gate $f$ on the input $x$ and on the hidden state $h$, this corresponds to having $f \in \left[\nu_0, \frac{1+\nu_0}{2}\right]$.

# B Theory

This section introduces all the theoretical results we directly or indirectly mention in the main text, as well as provides a proof for them.

## B.1 Useful lemmas

Most, if not all the calculations, that we will be doing in this section involves infinite sums. We state and prove two useful lemmas to simplify later calculations.

**Lemma 1.** *For $\alpha, \beta \in \mathbb{C}$ satisfying $|\alpha| < 1$ and $|\beta| < 1$, and $(u_n)_{n \in \mathbb{Z}}$ a bounded sequence satisfying $u_{-n} = u_n$, we have*

$$\sum_{n,m\geq 0} \alpha^n \beta^n u_{n-m} = \frac{1}{1-\alpha\beta}\left(u_0 + \sum_{\Delta \geq 1}(\alpha^\Delta + \beta^\Delta)u_\Delta\right) \tag{34}$$

*Proof.* The proof naturally comes from separating the indices $n$ and $m$ in three sets: one in which the two are equals, one in which $n$ is larger and one in which $m$ is larger. This gives

$$\sum_{n,m\geq 0} \alpha^n \beta^m u_{n-m} = \sum_{n=m} \alpha^n \beta^m u_{n-m} + \sum_{n>m} \alpha^n \beta^m u_{n-m} + \sum_{n<m} \alpha^n \beta^m u_{n-m} \tag{35}$$

$$= \sum_n \alpha^n \beta^n u_0 + \sum_m \alpha^m \beta^m \sum_{\Delta \geq 1} \alpha^\Delta u_\Delta + \sum_n \alpha^n \beta^n \sum_{\Delta \geq 1} \beta^\Delta u_{-\Delta} \tag{36}$$

$$= \sum_n \alpha^n \beta^n \left(u_0 + \sum_{\Delta \geq 1}(\alpha^\Delta + \beta^\Delta)u_\Delta\right) \tag{37}$$

$$= \frac{1}{1-\alpha\beta}\left(u_0 + \sum_{\Delta \geq 1}(\alpha^\Delta + \beta^\Delta)u_\Delta\right) \tag{38}$$

$\square$

**Lemma 2.** *In the same conditions as Lemma 1, we have*

$$\sum_{n,m\geq 0} nm\alpha^{n-1}\beta^{m-1}u_{n-m} = \frac{\mathrm{d}}{\mathrm{d}\alpha}\frac{\mathrm{d}}{\mathrm{d}\beta}\left[\frac{1}{1-\alpha\beta}\left(u_0 + \sum_{\Delta \geq 1}(\alpha^\Delta + \beta^\Delta)u_\Delta\right)\right] \tag{39}$$

*Proof.* This follows from remarking that

$$\frac{\mathrm{d}}{\mathrm{d}\alpha}\frac{\mathrm{d}}{\mathrm{d}\beta}\left[\sum_{n,m\geq 0} \alpha^n \beta^m u_{n-m}\right] = \frac{\mathrm{d}}{\mathrm{d}\alpha}\left[\sum_{n,m\geq 0} m\alpha^n \beta^{m-1}u_{n-m}\right] \tag{40}$$

$$= \sum_{n,m\geq 0} nm\alpha^{n-1}\beta^{m-1}u_{n-m} \tag{41}$$

and using Lemma 1 to get the final result. $\square$

## B.2 The curse of memory: signal propagation analysis

We recall the assumptions that we stated in Section 2.2:

a) **Linear diagonal recurrent neural networks**. We restrict ourselves to networks satisfying $h_{t+1} = \lambda \odot h_t + x_{t+1}$ with $\lambda$, $h_t$ and $x_t$ complex numbers. Without loss of generality, we focus on the one dimensional setting. We additionally consider $\lambda$s with absolute values smaller than 1.

b) **Infinite time horizon**. We consider infinite sequences and initialize the network dynamics at $t_0 = -\infty$.

c) **Wide-sense stationarity**. We assume the different quantities that the network receives, which includes the inputs $x_t$, to be wise-sense stationary (WSS). A random process $X_t$ is said to be WSS if its auto-correlation function is independent of time, that is, for all $t \in \mathbb{R}$ and $\Delta \in \mathbb{R}$, $\mathbb{E}\left[X_{t+\Delta} \bar{X}_t\right] := R_X(\Delta)$.

### B.2.1 Forward pass

Without loss of generality, we can take $t = 0$ given the wide-sense stationarity and infinite time horizon assumptions. Let us first remark that we have

$$h_0 = \sum_{n \geq 0} \lambda^n x_{-n} \tag{42}$$

so that

$$\mathbb{E}[|h_0|^2] = \sum_{n,m \geq 0} \lambda^n \bar{\lambda}^m \mathbb{E}\left[x_{-n} \bar{x}_{-m}\right] \tag{43}$$

$$= \sum_{n,m \geq 0} \lambda^n \bar{\lambda}^m R_x(n - m) \tag{44}$$

$$= \frac{1}{1 - |\lambda|^2} \left( R_x(0) + \sum_{\Delta \geq 1} (\bar{\lambda}^\Delta + \lambda^\Delta) R_x(\Delta) \right). \tag{45}$$

We used Lemma 1 to obtain the last equality. In Section 2.2, we focused on the real case $\bar{\lambda} = \lambda$, so this formula becomes Equation 5. If we further assume that the auto-correlation of $x$ decreases exponentially with decay rate $\rho$, that is $R_x(\Delta) = \rho^{|\Delta|}$, we can further simplify the last expression:

$$\mathbb{E}[|h_0|^2] = \frac{1}{1 - |\lambda|^2} \left( 1 + \sum_{\Delta \geq 1} (\bar{\lambda}^\Delta + \lambda^\Delta) \rho^\Delta \right) \tag{46}$$

$$= \frac{1}{1 - |\lambda|^2} \left( 1 + \frac{\bar{\lambda}\rho}{1 - \bar{\lambda}\rho} + \frac{\lambda\rho}{1 - \lambda\rho} \right) \tag{47}$$

$$= \frac{1 - \rho^2 |\lambda|^2}{|1 - \rho\lambda|^2 (1 - |\lambda|^2)} \tag{48}$$

It follows that if the inputs are i.i.d. ($\rho = 0$), we have $\mathbb{E}[|h_0|^2] = (1 - |\lambda|^2)^{-1}$, and if the inputs are constant equal to 1 ($\rho = 1$), we have $\mathbb{E}[|h_0|^2] = |1 - \lambda|^{-2}$.

### B.2.2 Backward pass

Differentiating the update $h_{t+1} = \lambda h_t + x_{t+1}$ with respect to $\lambda$ gives

$$\frac{\mathrm{d}h_{t+1}}{\mathrm{d}\lambda} = \lambda \frac{\mathrm{d}h_t}{\mathrm{d}\lambda} + h_t \tag{49}$$

so that

$$\frac{\mathrm{d}h_0}{\mathrm{d}\lambda} = \sum_{n \geq 0} \lambda^n h_{-n-1} \tag{50}$$

$$= \sum_{n \geq 0} \lambda^n \sum_{m \geq 0} \lambda^m x_{-n-m-1} \tag{51}$$

$$= \sum_{n,m \geq 0} \lambda^{n+m} x_{-(n+m+1)} \tag{52}$$

$$= \sum_{n \geq 0} n\lambda^{n-1} x_{-n-1}. \tag{53}$$

Note that some extra technicalities are needed to justify these equations as $\lambda$ and $h_t$ are complex valued: these formulas hold as they would in the real-valued case as $h_t$ is an holomorphic function of $\lambda$.

We can now compute the variance of the sensitivity of the hidden state with respect to the parameters.

$$\mathbb{E}\left[\left|\frac{\mathrm{d}h_t}{\mathrm{d}\lambda}\right|^2\right] = \sum_{n \geq 0} \sum_{m \geq 0} nm\lambda^{n-1}\bar{\lambda}^{m-1} R_x(n-m). \tag{54}$$

Using Lemma 2 gives

$$\mathbb{E}\left[\left|\frac{\mathrm{d}h_t}{\mathrm{d}\lambda}\right|^2\right] = \frac{\mathrm{d}}{\mathrm{d}\alpha}\frac{\mathrm{d}}{\mathrm{d}\beta}\left[\frac{1}{1-\alpha\beta}\left(R_x(0) + \sum_{\Delta \geq 1}(\alpha^\Delta + \beta^\Delta)R_x(\Delta)\right)\right]_{\alpha=\lambda,\beta=\bar{\lambda}}. \tag{55}$$

Differentiating this quantity as a product gives

$$\mathbb{E}\left[\left|\frac{\mathrm{d}h_t}{\mathrm{d}\lambda}\right|^2\right] = \left[\frac{1+\alpha\beta}{(1-\alpha\beta)^3}\left(R_x(0) + \sum_{\Delta \geq 1}(\alpha^\Delta + \beta^\Delta)R_x(\Delta)\right) + 0\right.$$
$$\left.+\frac{\alpha}{(1-\alpha\beta)^2}\left(\sum_{\Delta \geq 1}\Delta\alpha^{\Delta-1}R_x(\Delta)\right) + \frac{\beta}{(1-\alpha\beta)^2}\left(\sum_{\Delta \geq 1}\Delta\beta^{\Delta-1}R_x(\Delta)\right)\right]_{\alpha=\lambda,\beta=\bar{\lambda}}, \tag{56}$$

which then simplifies as

$$\mathbb{E}\left[\left|\frac{\mathrm{d}h_t}{\mathrm{d}\lambda}\right|^2\right] = \frac{1+|\lambda|^2}{(1-|\lambda|)^3}\left(R_x(0) + \sum_{\Delta \geq 1}(\lambda^\Delta + \bar{\lambda}^\Delta)R_x(\Delta)\right)$$
$$+\frac{1}{(1-|\lambda|^2)^2}\left(\sum_{\Delta \geq 1}\Delta(\lambda^\Delta + \bar{\lambda}^\Delta)R_x(\Delta)\right). \tag{57}$$

Note that Equation 6 in the main text is the real-valued version of that formula.

Let us now further simplify this equation when $R_x(\Delta) = \rho^{|\Delta|}$. If we use this in the differentiated quantity before differentiating it, we get

$$\mathbb{E}\left[\left|\frac{\mathrm{d}h_t}{\mathrm{d}\lambda}\right|^2\right] = \frac{\mathrm{d}}{\mathrm{d}\alpha}\frac{\mathrm{d}}{\mathrm{d}\beta}\left[\frac{1}{1-\alpha\beta}\left(\frac{1-\rho^2\alpha\beta}{(1-\rho\alpha)(1-\rho\beta)}\right)\right]_{\alpha=\lambda,\beta=\bar{\lambda}}. \tag{58}$$

Calculating this quantity manually is painful. Instead, we use the following trick. Its denominator is rather easy to compute, it is equal to $(1-\alpha\beta)^3(1-\rho\alpha)^2(1-\rho\beta)^2$. We thus multiply it to the derivative of the function we want to compute in order to obtain a polynomial with unknown factors, and use polynomial regression tools to derive the resulting coefficients. Massaging the obtained expression to make it easier to compute the closed-form value of this quantity when $\rho = 0$ and $\rho = 1$, we get

$$\mathbb{E}\left[\left|\frac{\mathrm{d}h_t}{\mathrm{d}\lambda}\right|^2\right] = \frac{(1-\rho)(1+|\lambda|^2) + \rho^2(1-|\lambda|^2)^3 + \rho(1-\rho)|\lambda|^2(\rho|\lambda|^2(1+|\lambda|^2) - 2\lambda - 2\bar{\lambda})}{(1-|\lambda|^2)^3|1-\rho\lambda|^4}. \tag{59}$$

This is the quantity we plot on Figure 1.A, when $\lambda$ is real-valued. When $\rho = 0$, this quantity becomes

$$\mathbb{E}\left[\left|\frac{\mathrm{d}h_t}{\mathrm{d}\lambda}\right|^2\right] = \frac{1+|\lambda|^2}{(1-|\lambda|^2)^3}, \tag{60}$$

and it is equal to

$$\mathbb{E}\left[\left|\frac{\mathrm{d}h_t}{\mathrm{d}\lambda}\right|^2\right] = \frac{1}{|1-\lambda|^4}, \tag{61}$$

when $\rho = 1$. Additionally, it will diverge whenever $|\lambda| \to 1$ when $\rho < 1$, and when $\lambda \to 1$ when $\rho = 1$.

Regarding the backpropagation of errors to the inputs, the analysis we did in the main text also holds for complex number given that $h_t$ is an holomorphic function of $x_t$ and it thus behaves as the forward pass once replacing the input distribution with the one of output errors $\partial_{h_t}L_t$.

### B.2.3 Extension to fully-connected networks

We now turn to the non-diagonal case. For the sake of simplicity, we assume that recurrent matrix is complex diagonalizable and that its eigenvalues are all different. This will enable us to differentiate the eigenvalues and the eigenvectors. We consider dynamics of the form

$$h_{t+1} = Ah_t + x_{t+1} \tag{62}$$

As $A$ is complex diagonalizable, there exists a complex-valued matrix $P$ and a complex-valued vector $\lambda$ such that

$$A = P\text{diag}(\lambda)P^{-1} \tag{63}$$

$$P_{:i}^{\dagger}P_{:i} = 1 \ \forall i. \tag{64}$$

The linear recurrent neural network considered above is equivalent to its diagonal version

$$h_{t+1}^{\text{diag}} = \lambda h_t^{\text{diag}} + P^{-1}x_{t+1} \tag{65}$$

$$h_t = Ph_t^{\text{diag}}. \tag{66}$$

We now differentiate $h_t$ w.r.t. to $A$ using the diagonal parametrization and obtain

$$\frac{\mathrm{d}h_t}{\mathrm{d}A} = \frac{\partial h_t}{\partial P}\frac{\partial P}{\partial A} + \frac{\partial h_t}{\partial h_t^{\text{diag}}}\frac{\mathrm{d}h_t^{\text{diag}}}{\mathrm{d}\lambda}\frac{\partial \lambda}{\partial A} + \frac{\partial h_t}{\partial h_t^{\text{diag}}}\frac{\mathrm{d}h_t^{\text{diag}}}{\mathrm{d}P^{-1}}\frac{\partial P^{-1}}{\partial A}. \tag{67}$$

$\mathrm{d}_A P$, $\mathrm{d}_A P^{-1}$ **and** $\mathrm{d}_A \lambda$ **can be considered constant.**  Intuitively, the eigenvalues and eigenvectors move smoothly as we restricted ourselves to the case in which eigenvalues are singular. If this is not the case, math becomes trickier as the eigenvectors are not uniquely defined. We can study the behavior of those quantities in more detail, following Boeddeker et al. [62]:

$$\frac{\partial \lambda}{\partial A_{ij}} = \text{diag}\left(P^{-1}\frac{\partial A}{\partial A_{ij}}P\right) \tag{68}$$

$$\frac{\mathrm{d}P}{\mathrm{d}A_{ij}} = P\left(F \odot \left(P^{-1}\frac{\partial A}{\partial A_{ij}}P\right)\right) \tag{69}$$

The $F$ introduced in the last equation is equal to

$$F_{ij} := \begin{cases} \frac{1}{\lambda_j - \lambda_i} & \text{if } i \neq j \\ 0 & \text{otherwise.} \end{cases} \tag{70}$$

Importantly, those two quantities do not grow to infinity as the absolute value of the eigenvalues goes to 1, which means that we can consider those derivatives to be independent of $|\lambda|$ for the sake of our analysis. Note that the previous argument assumes that eigenvalues do not collapse.

$\mathrm{d}_\lambda h_t^{\text{diag}}$ **is the dominating term in** $\mathrm{d}_A h_t$**.**  We wonder which of the three different terms that appear in $\mathrm{d}_A h_t$ (Equation 67) will be the dominating one as $|\lambda|$ (or $\lambda$) goes to 1. In the previous paragraph, we have shown that the derivative of $P^{-1}$, $P$ and $\lambda$ can be considered constant for the sake of our analysis. We thus focus on the other terms.

First, we have

$$\frac{\partial h_{t,l}}{\partial P_{ij}} = h_{t,i}^{\text{diag}}\mathbb{1}_{j=l} \tag{71}$$

so the magnitude of this quantity is roughly the one of $h_t^{\text{diag}}$, which corresponds to the low pass filtering of the inputs with different $\lambda$ values.

Second, we know that $\partial_{h_t^{\text{diag}}}h_t$ does not change in magnitude as $\lambda$ changes, as $P$ remains bounded.

So, for the third term of the sum, we are left to study the behavior of $\mathrm{d}_{P^{-1}}h_t^{\text{diag}}$. We can show that it evolves according to

$$\frac{\mathrm{d}h_{t+1,k}^{\text{diag}}}{\mathrm{d}P_{ij}^{-1}} = \lambda_i\frac{\mathrm{d}h_{t+1,k}^{\text{diag}}}{\mathrm{d}P_{ij}^{-1}} + x_{t+1,j} \text{ if } k = i \tag{72}$$

$$\frac{\mathrm{d}h_{t+1,k}^{\text{diag}}}{\mathrm{d}P_{ij}^{-1}} = 0 \text{ otherwise.} \tag{73}$$

It follows that the third term in the sum also corresponds to a low pass filtering of the inputs.

Finally, we know that the second term, the one in $d_\lambda h_t^{\mathrm{diag}}$ will grow faster to infinity as it corresponds to two consecutive low pass filters with the same $\lambda$ values (c.f. calculation above). It will thus be the dominating term in the infinite memory limit.

### B.2.4 On the wide-sense stationarity assumption

In our analysis, we make the assumption that the different quantities given to the network are wide-sense stationary, that is their statistics are invariant to a temporal shift. In practice, this assumption will likely never be satisfied, as sequences are finite and as parts of the sequence (e.g., the beginning of a text) can have different statistics than other parts (e.g., the end of a text).

It should be noted that the analysis we have done in the wide-sense stationary case can provide an upper bound on the different quantity we study. Indeed, if there exists a function $U$ such that

$$|\mathbb{E}[x_n \bar{x}_m]| \leq U(|n - m|), \tag{74}$$

minor modifications to our analysis enable to upper bound the different quantities we study, replacing $R_x$ by $U$.

In our experiments of Section 5, we plug the empirical covariance defined as

$$R_x^{\mathrm{empirical}}(\Delta) := \mathbb{E}_x \left[ \frac{1}{T - |\Delta| + 1} \sum_{t=0}^{T-|\Delta|} x_t x_{t+|\Delta|} \right] \tag{75}$$

into our analytical expressions. It comes with an approximation as it averages the autocorrelation for all positions, which are not equal when wide-sense stationarity is not met.

### B.3 Impact of input normalization and parametrization

In this section, we consider a diagonal linear recurrent neural network of the form

$$h_{t+1} = \lambda(\omega)h_t + \gamma(\lambda)x_{t+1} \tag{76}$$

with $\gamma(\lambda)$ the input normalization factor and $\lambda$ parametrized by a vector $\omega$. Next, we study the effect of input normalization and reparametrization, first in the real-valued setting and then in the complex-valued one.

### B.3.1 Real case

Let us start with the forward pass: as

$$h_t = \sum_{n \geq 0} \lambda^n \gamma(\lambda)x_{t-n}, \tag{77}$$

$\gamma$ rescales the value the hidden state takes. To avoid any explosion behavior, we thus ideally want $\gamma$ to be the inverse of value of $\mathbb{E}[(h_t)^2]$ without normalization, which we have computed in Equation 45. The same behavior holds for the backpropagation of errors to the inputs as

$$\frac{dL}{dx_t} = \gamma(\lambda) \left( \lambda \frac{dL}{dx_{t+1}} + \frac{\partial L}{\partial h_t} \right). \tag{78}$$

We now move to the impact of the parametrization. To simplify the calculation, we will ignore the dependency of $\gamma$ on $\lambda$ when differentiating it. This can easily be done in automatic differentiation software by removing this dependency from the computational graph with $\gamma(\mathrm{stop\_gradient}(\lambda))$. We then have

$$\frac{dh_t}{d\omega} = \frac{dh_t}{d\lambda} \frac{d\lambda}{d\omega} \tag{79}$$

and $d_\lambda h_t$ which is rescaled by $\gamma$ compared to the calculation we did above. As a consequence, both the input normalization and the parametrization can help to mitigate the curse of memory.

### B.3.2 On the difficulty of parametrizing complex numbers

We now extend the previous analysis to the complex case, and take a polar parametrization of $\lambda$: $\lambda(\omega) = \nu(\omega) \exp(i\theta(\omega))$. The effect of the input normalization does not change when moving to complex numbers. The role of the reparametrization is however a bit more subtle. As $h_t$ is an holomorphic function of $\lambda$, we have $d_{\bar{\lambda}} h_t = 0$ and

$$\frac{dh_t}{d\omega} = \frac{dh_t}{d\lambda} \frac{d\lambda}{d\omega} = \frac{dh_t}{d\lambda} \left( \frac{1}{2} \frac{d\nu}{d\omega} \exp(i\theta) + \frac{i}{2} \nu \frac{d\theta}{d\omega} \exp(i\theta) \right). \tag{80}$$

It follows that

$$\mathbb{E}\left[ \left| \frac{dh_t}{d\omega} \right|^2 \right] = \frac{1}{4} \mathbb{E}\left[ \left| \frac{dh_t}{d\lambda} \right|^2 \right] \left| \frac{d\nu}{d\omega} \exp(i\theta) + i\nu \frac{d\theta}{d\omega} \exp(i\theta) \right|^2 \tag{81}$$

$$= \frac{1}{4} \mathbb{E}\left[ \left| \frac{dh_t}{d\lambda} \right|^2 \right] \left| \frac{d\nu}{d\omega} + i\nu \frac{d\theta}{d\omega} \right|^2 \tag{82}$$

$$= \frac{1}{4} \mathbb{E}\left[ \left| \frac{dh_t}{d\lambda} \right|^2 \right] \left( \frac{d\nu}{d\omega}^2 + \nu^2 \frac{d\theta}{d\omega}^2 \right). \tag{83}$$

To simplify the analysis, we will further assume that $\mathbb{E}[|d_\lambda h_t|^2]$ is only a function of $\nu$. This asumption holds in the case of $\rho = 0$ and $\rho = 1$, c.f. Section B.2.2, but not necessarily otherwise. To ensure that that this quantity does not depend on $\lambda$, we thus want $d_\omega \nu^2 E(\nu) = \Theta(1)$ and $\nu^2 d_\omega \theta^2 E(\nu) = \Theta(1)$. The second means that the angle parametrization must depend on the value $\nu$ takes. Let us take the $\rho = 0$ example to get an idea of what the ideal parametrization should be. First, we have $\gamma(\lambda) = \sqrt{1 - \nu^2}$ so that

$$E(\nu) = \gamma(\lambda)^2 \frac{1 + \nu^2}{(1 - \nu^2)^3} = \frac{1 + \nu^2}{(1 - \nu^2)^2}. \tag{84}$$

We are left with the differential equation $\nu' = \Theta(1 - \nu^2)$, which is for example solved with $\nu = \tanh(\omega_\nu)$. Now let us look at the parametrization of $\theta$. If we ignore the $\nu^2$ term for simplicity, the approximate differential equation it needs to solve is $d_\omega \theta = \Theta(1 - \nu^2)$, which can be solved by $\theta = \text{stop\_gradient}(1 - \nu^2)\omega_\theta$. The exact detail of this calculation do not really matter as this is heavily input distribution dependent. However, the interesting part here is that the angle parameter must be rescaled by a function of $\nu$. This makes intuitive sense when looking looking at the sharpness of the loss around optimality in Figure 1.C, but this also makes the loss even flatter further away from optimality. We will come back to this point in Section C.1.4, showing that in practice, such a parametrization complicates the learning of the $\theta$. Learning complex numbers is thus difficulty, because of the angle.

# C Linear teacher-student task

This section is dedicated to detail the theoretical results behind our analysis of the teacher-student task, present all the details necessary to reproduce our empirical experiments, and provide additional analysis.

## C.1 1D setting

### C.1.1 Calculation of the loss

In this toy example, we are interested in learning a simple 1-dimensional linear recurrent neural network which follows the dynamics

$$h_{t+1} = \lambda h_t + x_{t+1} \tag{85}$$

to reproduce the hidden state $h_t^*$ of a teacher with recurrent parameter $\lambda^*$. Note that we here allow all variables to be complex-valued. We take the loss to be

$$L(\lambda, \lambda^*) := \frac{1}{2T} \sum_{t=1}^{T} \mathbb{E}_x \left[ |h_t - h_t^*|^2 \right] \tag{86}$$

We assume $x$ to be drawn from a wide-sense stationary distribution so that we can focus on studying the behavior of one $L_t(\lambda, \lambda^*) := \frac{1}{2}\mathbb{E}_x \left[ |h_t - h_t^*|^2 \right]$ to understand the behavior of the full loss $L$, in the limit of infinitely long sequences ($T \to \infty$). Moreover, to further simplify the calculations, we assume that $x$ is real-valued and that $R_x(\Delta) = \rho^{|\Delta|}$.

Let us now proceed with the calculation:

$$L_t(\lambda, \lambda^*) := \frac{1}{2}\mathbb{E}_x \left[ h_t \bar{h}_t + h_t^* \bar{h}_t^* - h_t \bar{h}_t^* - \bar{h}_t h_t^* \right]. \tag{87}$$

We have shown in Section B.2 that in the limit of $t \to \infty$,

$$\mathbb{E}_x \left[ h_t \bar{h}_t \right] = \frac{1}{1 - \lambda\bar{\lambda}} \left( 1 + \frac{\rho\lambda}{1 - \rho\lambda} + \frac{\rho\bar{\lambda}}{1 - \rho\bar{\lambda}} \right) \tag{88}$$

$$\tag{89}$$

Similar derivations hold for the other three terms in the loss. Grouping them gives the exact value of the loss. We omit the formula as it is not particularly insightful. In the case of constant inputs ($\rho = 1$), we have

$$L_t(\lambda, \lambda^*) = \frac{1}{2} \left| \frac{1}{1 - \lambda} - \frac{1}{1 - \lambda^*} \right|^2. \tag{90}$$

In the case of i.i.d. inputs ($\rho = 0$), we have

$$L_t(\lambda, \lambda^*) = \frac{1}{2} \left( \frac{1}{1 - |\lambda|^2} + \frac{1}{1 - |\lambda^*|^2} - \text{Re} \left[ \frac{2}{1 - \bar{\lambda}\lambda^*} \right] \right). \tag{91}$$

This is the loss we plot on Figure 1.B and C.

### C.1.2 Optimal normalization and reparametrization with uncorrelated inputs

Having a simple closed-form solution for the value the loss takes gives us the possibility to investigate in more detail what an optimal normalization and parametrization should be. We focus on the case $\rho = 0$.

For $\rho = 0$, the optimal normalization is $\gamma(\lambda) = \sqrt{1 - |\lambda|^2}$. Given that we now add an input normalization to the student, we must also add it to the teacher for the student to be able to fit it. The loss becomes

$$L_t = \frac{1}{2} \left( \frac{\gamma(\lambda)}{1 - |\lambda|^2} + \frac{\gamma(\lambda^*)}{1 - |\lambda^*|^2} - \text{Re} \left[ \frac{2\gamma(\lambda)\gamma(\lambda^*)}{1 - \bar{\lambda}\lambda^*} \right] \right) \tag{92}$$

$$= 1 - \text{Re} \left[ \frac{\gamma(\lambda)\gamma(\lambda^*)}{1 - \bar{\lambda}\lambda^*} \right]. \tag{93}$$

Next, we parametrize $\lambda$ as $\lambda = \nu(\omega_\nu)\exp(i\theta(\omega_\theta))$ and seek to find a parametrization such that, at optimality, $\mathbb{E}[(\mathrm{d}_{\omega_\nu} h_t)^2] = 1$ and $\mathbb{E}[(\mathrm{d}_{\omega_\theta} h_t)^2] = 1$. Given that the student perfectly fit the teacher-generated data at optimality and that the loss we use is the mean-squared error, this corresponds to having $\mathrm{d}^2_{\omega_\nu} L_t = 1$ and $\mathrm{d}^2_{\omega_\theta} L_t = 1$.

**Deriving the optimal $\nu$ parametrization.** We now compute the Hessian of the loss w.r.t. $\omega_\nu$. First, we can simplify our calculations by restricting ourselves to the case $\theta = \theta^*$. The loss becomes

$$L_t = 1 - \frac{\gamma(\nu)\gamma(\nu^*)}{1 - \nu\nu^*}. \tag{94}$$

Differentiating this function a first time, we obtain

$$\frac{\mathrm{d}L_t}{\mathrm{d}\nu} = -\frac{\gamma(\nu^*)\gamma'(\nu)}{1 - \nu\nu^*} - \frac{\gamma(\nu^*)\nu^*\gamma(\nu)}{(1 - \nu\nu^*)^2}. \tag{95}$$

Differentiating it a second time gives

$$\frac{\mathrm{d}^2 L_t}{\mathrm{d}\nu^2} = -\frac{\gamma(\nu^*)\gamma''(\nu)}{1 - \nu\nu^*} - \frac{2\gamma(\nu^*)\nu^*\gamma'(\nu)}{(1 - \nu\nu^*)^2} - \frac{2\gamma(\nu^*)\nu^{*2}\gamma(\nu)}{(1 - \nu\nu^*)^3}. \tag{96}$$

Leveraging the fact that

$$\gamma'(\nu) = \frac{-\nu}{\gamma(\nu)} \quad \text{and} \quad \gamma''(\nu) = \frac{-\gamma(\nu)^2 - \nu^2}{\gamma(\nu)^3}, \tag{97}$$

we finally get, when $\nu = \nu^*$,

$$\frac{\mathrm{d}^2 L_t}{\mathrm{d}\nu^2} = \frac{1}{(1 - \nu^2)^2}. \tag{98}$$

Given that we are at optimality, we have

$$\frac{\mathrm{d}^2 L_t}{\mathrm{d}\omega_\nu^2} = \mathbb{E}\left[\frac{\mathrm{d}h_t}{\mathrm{d}\omega_\nu}\frac{\mathrm{d}^2 L_t}{\mathrm{d}h_t^2}\frac{\mathrm{d}h_t}{\mathrm{d}\omega_\nu}\right] = \nu'(\omega_\nu)^2 \mathbb{E}\left[\frac{\mathrm{d}h_t}{\mathrm{d}\nu}\frac{\mathrm{d}^2 L_t}{\mathrm{d}h_t^2}\frac{\mathrm{d}h_t}{\mathrm{d}\nu}\right] = \nu'(\omega_\nu)^2 \frac{\mathrm{d}^2 L_t}{\mathrm{d}\nu^2}. \tag{99}$$

To keep that quantity constant, we thus have to solve the differential equation

$$\nu'(\omega_\nu) = (1 - \nu^2), \tag{100}$$

which gives $\nu(\omega_\nu) = \tanh(\omega_\nu)$.

**Deriving the optimal $\theta$ parametrization.** We now move to the parametrization of $\theta$. We have

$$L_t = 1 - \mathrm{Re}\left[\frac{\gamma(\nu)\gamma(\nu^*)}{1 - \bar{\lambda}\lambda^*}\right] = 1 - \frac{\gamma(\nu)\gamma(\nu^*)(1 - \nu\nu^*\cos(\theta - \theta^*))}{|1 - \bar{\lambda}\lambda^*|^2}. \tag{101}$$

For notational convenience, we denote

$$\alpha(\theta - \theta^*) := |1 - \bar{\lambda}\lambda^*|^2 = (1 - \nu\nu^*\cos(\theta - \theta^*))^2 + \nu^2\nu^{*2}\sin(\theta - \theta^*)^2. \tag{102}$$

We have

$$\frac{\mathrm{d}L_t}{\mathrm{d}\theta} = \gamma(\nu)\gamma(\nu^*)\left(-\frac{\nu\nu^*\sin(\theta - \theta^*)}{\alpha(\theta - \theta^*)} + \frac{(1 - \nu\nu^*\cos(\theta - \theta^*))\alpha'(\theta - \theta^*)}{\alpha(\theta - \theta^*)^2}\right) \tag{103}$$

and

$$\begin{aligned}
\frac{\mathrm{d}^2 L_t}{\mathrm{d}\theta^2} = \gamma(\nu)\gamma(\nu^*)\Bigg(&-\frac{\nu\nu^*\cos(\theta - \theta^*)}{\alpha(\theta - \theta^*)} + 2\frac{\nu\nu^*\sin(\theta - \theta^*)\alpha'(\theta - \theta^*)}{\alpha(\theta - \theta^*)^2} \\
&+\frac{(1 - \nu\nu^*\cos(\theta - \theta^*))\alpha''(\theta - \theta^*)}{\alpha(\theta - \theta^*)^2} - 2\frac{(1 - \nu\nu^*\cos(\theta - \theta^*))\alpha'(\theta - \theta^*)^2}{\alpha(\theta - \theta^*)^3}\Bigg)
\end{aligned} \tag{104}$$

At optimality ($\theta = \theta^*$ and $\nu = \nu^*$), we have $\alpha(0) = (1 - \nu^2)^2$, $\alpha'(0) = 0$ and $\alpha''(0) = 2\nu^2$, so that

$$\frac{\mathrm{d}^2 L_t}{\mathrm{d}\theta^2} = \frac{\nu^2(1 + \nu^2)}{(1 - \nu^2)^2}. \tag{105}$$

The optimal parametrization thus has to satisfy

$$\theta'(\omega_\theta) = \frac{1 - \nu^2}{\nu\sqrt{1 + \nu^2}}, \tag{106}$$

that is

$$\theta(\omega_\theta) = \omega_\theta \frac{1 - \nu^2}{\nu\sqrt{1 + \nu^2}} \tag{107}$$

There are two things we can remark:

- First, the parametrization that we derived for the general case in Section B.3.2, which additionally ignored the dependence of $\gamma$ on $\lambda$, is relatively accurate. The only difference is the apparition of the extra $\nu$ term, which becomes insignificant in the long memory limit $\nu \to 1$.

- Second, the optimal $\theta$ parametrization has to be a function of $\nu$, and thus $\omega_\nu$, so the differential equation $\nu$ needs to satisfy changes. Yet, this considerably simplifies the calculation and there is no simple solution to that problem. One could still argue that the initial choice we made, that is to use a polar parametrization, is the issue. It could be, but most practical models end up using that choice so highlighting the limitations of this choice has important practical consequences.

In the rest of this section, we ignore the dependency of $\theta$ on $\nu$, and consider the optimal parametrization in this setting to be

$$\nu(\omega_\nu^{\mathrm{opt}}) = \tanh(\omega_\nu^{\mathrm{opt}}) \tag{108}$$

$$\theta(\omega_\theta^{\mathrm{opt}}) = \omega_\theta^{\mathrm{opt}} \frac{1 - \nu^2}{\nu\sqrt{1 + \nu^2}}. \tag{109}$$

### C.1.3 Visualization of the effect of input normalization and reparametrization

We now visualize the effect of input normalization and reparametrization on the loss landscape. We focus on two such reparametrizations:

- the one used in the LRU [20, 22] with $\gamma(\lambda) = \sqrt{1 - \lambda^2}$, $\nu = \exp(-\exp(\omega_\nu^{\mathrm{exp}}))$ and $\theta = \exp(\omega_\theta^{\mathrm{exp}})$.

- the optimal one we derived in the previous Section (c.f. Equations 108 and 109), which is tailored to this specific setting.

### C.1.4 Learning the angle is difficult in practice: an example

We use this one-dimensional teacher-student setting to test whether having a parametrization that avoids exploding behaviors at optimality, such as the one we derived in Section C.1.2, facilitates learning. Figure 6 already hints towards the fact the basin of attraction of the global minima is either extremely narrow or that their number decreases as longer memories are considered, making learning more tedious. Figure 7 confirms it. In this figure, we plot the learning dynamics obtained using the Adam optimizer with a learning rate of $10^{-3}$ for 50k steps, starting from $\lambda_0 = 0.99\exp(i\pi/4)$. We consider three different parametrizations of the angle:

$$\theta(\omega_\theta^{\mathrm{polar}}) = \omega_\theta^{\mathrm{polar}} \tag{110}$$

$$\theta(\omega_\theta^{\mathrm{exp}}) = \log(\omega_\theta^{\mathrm{exp}}) \tag{111}$$

$$\theta(\omega_\theta^{\mathrm{opt}}) = \frac{(1 - \nu^2)}{\nu\sqrt{1 + \nu^2}}\omega_\theta. \tag{112}$$

The first one does not reparametrize the angle, the second one is the one used in the LRU and the third one is the optimal one we derived above. We use $\nu = \tanh(\omega_\nu)$ to parametrize the magnitude in the three cases. We set $\lambda^*$ to $\lambda^* = 0.99\exp(i\pi/100)$. The $\theta$ landscape when $\nu$ is correct therefore corresponds to the ones plotted in the last two columns of Figure 6. This example shows that efforts to reduce the sharpness of the loss at optimality, as done in the last parametrization, inevitably make the loss flatter elsewhere and optimization impossible.

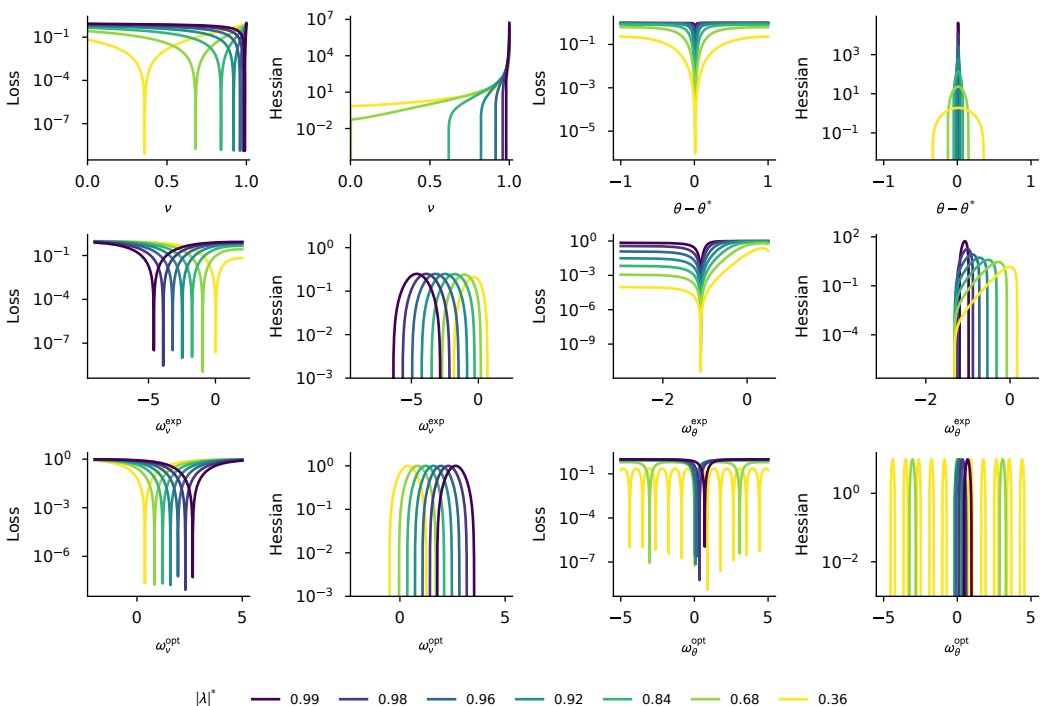

**Figure 6: Visualization of the loss landscape with input normalization, in the teacher and the student, for different parametrizations.** The teacher satisfies $\lambda^* = |\lambda^*| \exp(i\pi/100)$, for different $|\lambda^*|$ values. The first two columns correspond to students with correct angle $\theta = \theta^*$ but wrong absolute value $\nu$ and the last two columns to students with correct absolute value $\nu = |\lambda^*|$ but wrong angle. When we fix one variable, we ignore how it affects the loss for the Hessian caclulation. Each line corresponds to a different parametrization: the first line uses a polar parametrization ($\lambda = \nu \exp(i\theta)$), the second line uses the double exponential parametrization used in the LRU (exp) and the third one is the optimal parametrization for that task (tanh). Overall, both reparametrizations enable to control the explosion of the Hessian. However, the size of basins of attraction around optimality, or their number, shrinks as $|\lambda^*|$ goes to 1 for the angle, but not for the absolute value, highlighting how difficult learning the angle can be.

## C.2  Structure of the Hessian at optimality

In Section 4, we argue that the Hessian at optimality is an important object to understand the learning dynamics in the linear teacher-student task we consider. We here provide some theoretical analysis of its structure in the complex diagonal setting, that is we consider a recurrent network of the form

$$h_{t+1} = \lambda \odot h_t + bx_t \tag{113}$$

$$y_t = \mathrm{Re}[c^\top h_t] + dx_t. \tag{114}$$

with $\lambda$, $b$ and $c$ complex vectors of size $n$, with $n$ the number of hidden neurons, and $d$ a scalar. We additionally take the loss to be the mean-square error, which is also the one we use in our numerical experiments. Note that, as in our theoretical analysis of Section 2, we consider infinitely long sequences and wide-sense stationary inputs.

Recall that the Hessian of the loss is equal to

$$\frac{\mathrm{d}^2 L}{\mathrm{d}\theta^2} = \sum_t \mathbb{E}_x \left[ \frac{\mathrm{d}h_t}{\mathrm{d}\theta} \frac{\partial^2 L_t}{\partial h_t^2} \frac{\mathrm{d}h_t}{\mathrm{d}\theta}^\top + \frac{\partial L_t}{\partial h_t} \frac{\mathrm{d}^2 h_t}{\mathrm{d}\theta^2} \right]. \tag{115}$$

At optimality, only the first term remains, as $\partial_{h_t} L_t$ is 0 for all data points. Given that we have shown earlier, e.g. in Section B.2, that the most sensitive parameters to learn are the recurrent ones $\lambda$, we focus on the Hessian with respect to these parameters in the following.

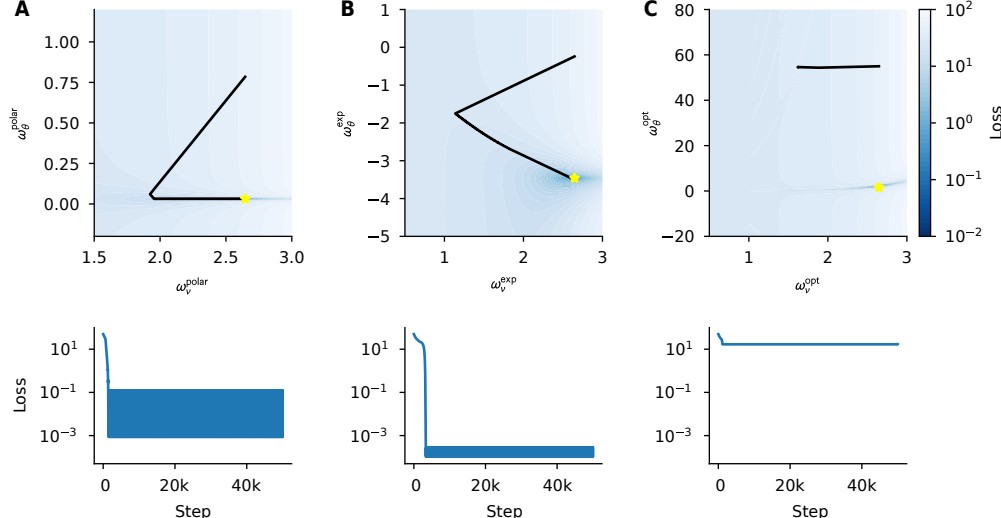

Figure 7: **Learning the angle is difficult, even in a simple one-dimensional task.** The target $\lambda$ value is equal to $\lambda^* = 0.99 \exp(i\pi/100)$ and is plotted in yellow. The black lines correspond to the Adam learning dynamics. **A.** When the angle is not reparametrized ($\theta = \omega_\theta$), the loss landscape is extremely sharp in the $\omega_\theta$ direction, but Adam compensates for it. **B.** When the angle is parametrized exponentially ($\theta = \exp(\omega_\theta)$), the loss landscape becomes smoother. However, this only hold when the considered angles are small enough, as the exponential parametrization does not bring extra granularity elsewhere. **C.** When reparametrizing the angle to reduce the gradient explosion as $|\lambda| \to 1$, the loss becomes extremely tricky to navigate. The parameters are first attracted to a nearby valley, which is flat on the $\omega_\theta$ direction and only indirectly connected to the global minimum. Such a reparametrization thus hinders optimization far away from optimality. See Section C.1.4 for more detail.

### C.2.1  Hessian for complex-valued variables

Before delving into more specific calculations, we make a few remarks on how to deal the Hessian when having complex-valued parameters. We will mostly leverage the fact that the loss $L$ is real-valued.

Before that, we recall a few facts about Wirtinger derivatives:

–  For $f(z)$ a complex-valued function of $z$, the Wirtinger derivatives are defined as:

$$\frac{\mathrm{d}f}{\mathrm{d}z} = \frac{1}{2}\left(\frac{\mathrm{d}\mathrm{Re}[f]}{\mathrm{d}\mathrm{Re}[z]} - i\frac{\mathrm{d}\mathrm{Im}[f]}{\mathrm{d}\mathrm{Im}[z]}\right) \tag{116}$$

$$\frac{\mathrm{d}f}{\mathrm{d}\bar{z}} = \frac{1}{2}\left(\frac{\mathrm{d}\mathrm{Re}[f]}{\mathrm{d}\mathrm{Re}[z]} + i\frac{\mathrm{d}\mathrm{Im}[f]}{\mathrm{d}\mathrm{Im}[z]}\right). \tag{117}$$

–  We have

$$\frac{\mathrm{d}f}{\mathrm{d}z} = \overline{\frac{\mathrm{d}f}{\mathrm{d}\bar{z}}}. \tag{118}$$

–  Leveraging the fact that $L$ is real-valued so that $\bar{L} = L$, we have

$$\frac{\mathrm{d}^2 L}{\mathrm{d}\lambda^2} = \frac{\mathrm{d}}{\mathrm{d}\lambda}\left[\frac{\mathrm{d}L}{\mathrm{d}\lambda}^\top\right] \tag{119}$$

$$= \frac{\mathrm{d}}{\mathrm{d}\lambda}\left[\overline{\frac{\mathrm{d}L}{\mathrm{d}\bar{\lambda}}}^\top\right] \tag{120}$$

$$= \overline{\frac{\mathrm{d}^2 L}{\mathrm{d}\bar{\lambda}^2}} \tag{121}$$

and, similarly, $d_\lambda d_{\bar\lambda} L = \overline{d_{\bar\lambda} d_\lambda L}$. Second derivatives are symmetric, so we additionally have $d_\lambda d_{\bar\lambda} L = d_{\bar\lambda} d_\lambda L^\top$, which means that the complex Hessian is a Hermitian matrix.

Taken all together, this shows that the full complex Hessian, which contains all cross derivatives, has a similar structure to the real case.

### C.2.2 Hessian with respect to the recurrent eigenvalues

In this section, we compute the full complex Hessian with respect to the recurrent eigenvalue $\lambda$ and defer the analysis of reparametrization to the next section.

First, let us remark that

$$\frac{dL_t}{d\lambda} = \frac{\partial L_t}{\partial y_t} c^\top \frac{dh_t}{d\lambda} \tag{122}$$

$$\tag{123}$$

so that

$$\frac{d^2 L_t}{d\lambda^2} = \frac{d}{d\lambda}\left[\frac{dh_t}{d\lambda}^\top c \frac{\partial L_t}{\partial y_t}^\top\right] \tag{124}$$

$$= \frac{d^2 h_t}{d\lambda^2} c \frac{\partial L_t}{\partial y_t}^\top + \frac{dh_t}{d\lambda}^\top c \frac{\partial^2 L_t}{\partial y_t^2} c^\top \frac{dh_t}{d\lambda} \tag{125}$$

We assumed that we are at optimality so that the network perfectly fits the target trajectories and $\partial_{y_t} L_t = 0$. Additionally, $L_t$ is the mean-squared error loss so that $\partial_{y_t}^2 L_t = \text{Id}$. It follows that

$$\left(\frac{d^2 L_t}{d\lambda^2}\right)_{ij} = \left(\frac{dh_t}{d\lambda}^\top cc^\top \frac{dh_t}{d\lambda}\right)_{ij} \tag{126}$$

$$= \left(c^\top \frac{dh_t}{d\lambda}\right)_i \left(c^\top \frac{dh_t}{d\lambda}\right)_j \tag{127}$$

$$= c_i \frac{dh_{t,i}}{d\lambda_i} c_j \frac{dh_{t,j}}{d\lambda_j}. \tag{128}$$

In the last equation, we made use of the fact that the parameter $\lambda_i$ only affects the hidden state $h_{t,i}$ and not the others, so $d_{\lambda_j} h_{t,i} = 0$ if $i \neq j$.

The previous calculation applied to one sequence, we now take the expectation over the data:

$$\frac{d^2 L}{d\lambda^2} = (cc^\top) \odot \mathbb{E}_{x,y}\left[\lim_{t\to\infty} \frac{dh_t}{d\lambda} \frac{dh_t}{d\lambda}^\top\right] \tag{129}$$

Note that we introduced a slight abuse of notation in the previous equation as $d_\lambda h_t$ is in general a matrix. However, given that the hidden neurons are independent here due to the diagonal connectivity, it is effectively a vector, and we treat it that way. Let us now compute the expectation, using similar calculation techniques to the one we used in Section B.2:

$$\mathbb{E}_{x,y}\left[\lim_{t\to\infty} \frac{dh_{t,i}}{d\lambda_i} \frac{dh_{t,j}}{d\lambda_j}\right] = \mathbb{E}\left[\sum_{n,m\geq 0} n\lambda_i^{n-1} b_i x_{-n} m\lambda_j^{m-1} b_j x_{-m}\right] \tag{130}$$

$$= b_i b_j \mathbb{E}\left[\sum_{n,m\geq 0} n\lambda_i^{n-1} x_{-n} m\lambda_j^{m-1} x_{-m}\right] \tag{131}$$

$$= b_i b_j \sum_{n,m\geq 0} nm\lambda_i^{n-1} \lambda_j^{m-1} R_x(n-m) \tag{132}$$

We can now remark that this quantity is very similar to the one we have encountered in Section B.2, up to the presence of $b_i b_j$, and can be simplified using Lemma 2. For conciseness, we note $S(\lambda_i, \lambda_j)$

the right-hand side of the last equation without the $b_i b_j$ factor. Putting this result back in the Hessian, we get

$$\frac{\mathrm{d}^2 L}{\mathrm{d}\lambda_i \mathrm{d}\lambda_j} = b_i b_j c_i c_j S(\lambda_i, \lambda_j) \tag{133}$$

To gain further intuition of the behavior of this quantity, we take $R_x(\Delta) = \rho^{|\Delta|}$, $\rho$ being a real number. A similar calculation to the one we did in Section B.2 gives

$$S(\lambda_i, \lambda_j) = \frac{(1-\rho)(1+\lambda_i\lambda_j) + \rho^2(1-\lambda_i\lambda_j)^3 + \rho(1-\rho)\lambda_i\lambda_j(\rho\lambda_i\lambda_j(1+\lambda_i\lambda_j) - 2\lambda_i - 2\lambda_j)}{(1-\lambda_i\lambda_j)^3(1-\rho\lambda_i)^2(1-\rho\lambda_j)^2}. \tag{134}$$

Given the complexity of this formula, we visualize the magnitude of $S(\lambda_i, \lambda_j)$ on Figure 8. Interestingly, we observe this quantity is large when $\lambda_i$ and $\lambda_j$ are conjugate to each other and inputs are uncorrelated. However, as elements in the input sequence get more correlated ($\rho \to 1$), this effect disappears and $|S|$ increases as one of the two eigenvalue gets closer to 1 in the complex plane. In both cases, the effect gets amplified as the magnitude of the eigenvalue increases.

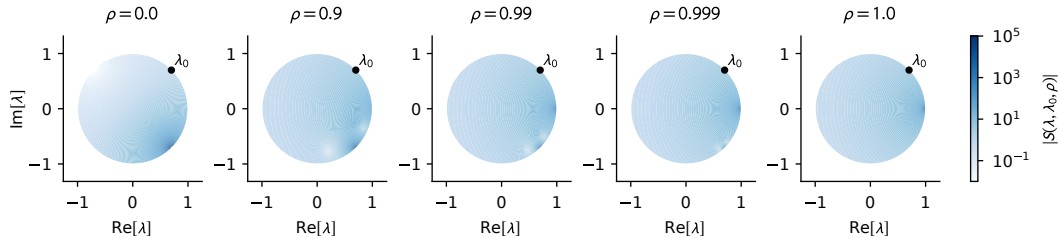

Figure 8: **Visualization of** $\lambda \mapsto |S(\lambda, \lambda_0)|$ **for** $\lambda_0 = 0.99\exp(i\pi/4)$**.** This term measures how "similar" eigenvalues are in the Hessian. When $\rho = 0$, eigenvalues are mostly "similar" when they are conjugate to each other. As $\rho$ increases, this effect decreases and eigenvalues become more "similar" if one of them gets close to 1.

We also need to compute $\mathrm{d}_{\bar{\lambda}}\mathrm{d}_\lambda L$ to get the full complex Hessian. Similarly to the previous calculation, we first have

$$\frac{\mathrm{d}L_t}{\mathrm{d}\bar{\lambda}} = \overline{\frac{\mathrm{d}\bar{L}_t}{\mathrm{d}\lambda}} = \overline{\frac{\mathrm{d}\bar{L}_t}{\mathrm{d}\lambda}} = \frac{\partial L_t}{\partial y_t} \bar{c}^\top \overline{\frac{\mathrm{d}h_t}{\mathrm{d}\lambda}}. \tag{135}$$

It follows that

$$\frac{\mathrm{d}^2 L}{\mathrm{d}\lambda_i \mathrm{d}\bar{\lambda}_j} = \mathbb{E}\left[\overline{\frac{\mathrm{d}h_{t,j}}{\mathrm{d}\lambda_j}} \bar{c}_j c_i \frac{\mathrm{d}h_{t,i}}{\mathrm{d}\lambda_i}\right] \tag{136}$$

$$= c_i \bar{c}_j b_i \bar{b}_j S(\lambda_i, \bar{\lambda}_j). \tag{137}$$

Using the symmetry with the complex Hessian matrix, we now have all its components.

### C.2.3 Hessian for different parametrizations

So far, we have computed the complex Hessian, which is not of direct use as we end up optimizing real numbers in practice. Here, we study the impact of different parametrizations of $\lambda$ on the Hessian. Given that this parametrization only affects $\lambda$ and not the other parameters in the network and that we only consider the Hessian at optimality here, computing the Hessian of those parameters reduces to left and right multiplying the Hessian by derivatives of $\lambda$ and $\bar{\lambda}$ with respect to these parameters. For future reference, we introduce

$$H_{ij}^\lambda := \begin{pmatrix} \frac{\mathrm{d}^2 L}{\mathrm{d}\lambda_i \mathrm{d}\lambda_j} & \frac{\mathrm{d}^2 L}{\mathrm{d}\lambda_i \mathrm{d}\bar{\lambda}_j} \\ \frac{\mathrm{d}^2 L}{\mathrm{d}\bar{\lambda}_i \mathrm{d}\lambda_j} & \frac{\mathrm{d}^2 L}{\mathrm{d}\bar{\lambda}_i \mathrm{d}\bar{\lambda}_j} \end{pmatrix} = \begin{pmatrix} A_{ij} & B_{ij} \\ \bar{B}_{ij} & \bar{A}_{ij}. \end{pmatrix} \tag{138}$$

with $A_{ij} := b_i b_j c_i c_j S(\lambda_i, \lambda_j)$ and $B_{ij} = b_i \bar{b}_j c_i \bar{c}_j S(\lambda_i, \bar{\lambda}_j)$.

**Real-imaginary parametrization:** $\lambda = \omega_{\mathrm{re}} + \omega_{\mathrm{im}}$. We aim at computing the matrix

$$H_{ij}^{\mathrm{RI}} := \begin{pmatrix} \frac{\mathrm{d}^2 L}{\mathrm{d}\omega_{\mathrm{re},i}\mathrm{d}\omega_{\mathrm{re},j}} & \frac{\mathrm{d}^2 L}{\mathrm{d}\omega_{\mathrm{re},i}\mathrm{d}\omega_{\mathrm{im},j}} \\ \frac{\mathrm{d}^2 L}{\mathrm{d}\omega_{\mathrm{im},i}\mathrm{d}\omega_{\mathrm{re},j}} & \frac{\mathrm{d}^2 L}{\mathrm{d}\omega_{\mathrm{im},i}\mathrm{d}\omega_{\mathrm{im},j}} \end{pmatrix}, \tag{139}$$

which is the building block to compute the full Hessian. First, let us remark that $\mathrm{d}_{\omega_{\mathrm{re},i}}\lambda_i = 1/2$, $\mathrm{d}_{\omega_{\mathrm{re},i}}\bar{\lambda}_i = 1/2$, $\mathrm{d}_{\omega_{\mathrm{im},i}}\lambda_i = i/2$ and $\mathrm{d}_{\omega_{\mathrm{im},i}}\bar{\lambda}_i = -i/2$. It follows that

$$\frac{\mathrm{d}^2 L}{\mathrm{d}\omega_{\mathrm{re},i}\mathrm{d}\omega_{\mathrm{re},j}} = (\mathrm{d}_{\omega_{\mathrm{re},j}}\lambda_j \ \ \mathrm{d}_{\omega_{\mathrm{re},j}}\bar{\lambda}_j)H_{ij}^{\lambda}(\mathrm{d}_{\omega_{\mathrm{re},i}}\lambda_i \ \ \mathrm{d}_{\omega_{\mathrm{re},i}}\bar{\lambda}_i)^{\top} \tag{140}$$

$$= \frac{1}{4}(1\ \ 1)H_{ij}^{\lambda}(1\ \ 1)^{\top} \tag{141}$$

$$= \frac{1}{2}\left(\mathrm{Re}[A_{ij}] + \mathrm{Re}[B_{ij}]\right). \tag{142}$$

Once again we emphasize that the first line only holds as we are at optimality. Similar calculations give the rest of the elements of $H_{ij}^{\mathrm{RI}}$:

$$H_{ij}^{\mathrm{RI}} := \frac{1}{2}\begin{pmatrix} \mathrm{Re}[A_{ij} + B_{ij}] & \mathrm{Im}[-A_{ij} + B_{ij}] \\ \mathrm{Im}[-A_{ij} - B_{ij}] & \mathrm{Re}[-A_{ij} + B_{ij}]. \end{pmatrix}. \tag{143}$$

Given the intuition we gained on the structure of $S$ previously, and the fact that $A_{ij} \propto S(\lambda_i, \lambda_j)$ and $B_{ij} \propto S(\lambda_i, \bar{\lambda}_j)$, we know that this block will have large components if the two corresponding eigenvalues are conjugate of each other or aligned to each other, or if one of them is close to 1.

One other quantity that we can calculate is the trace of the Hessian $H^{\mathrm{RI}}$, which is equal to the sum of its eigenvalues. Note that this does not correspond to the eigenvalues of the full Hessian matrix, as it additionally contains entries for other parameters. Yet it already provides some idea of how large the Hessian will be, as the value of this trace appears in the value of the full trace. We have

$$\mathrm{Tr}[H^{\mathrm{RI}}] = \sum_i \frac{1}{2}\left(\mathrm{Re}[A_{ii} + B_{ii}] + \mathrm{Re}[-A_{ii} + B_{ii}]\right) \tag{144}$$

$$= \sum_i \mathrm{Re}[B_{ii}] \tag{145}$$

$$= \sum_i |b_i|^2 |c_i|^2 S(\lambda_i, \bar{\lambda}_i) \tag{146}$$

where we used that $S(\lambda_i, \bar{\lambda}_i)$ is real-valued in the last line. As a side note, this formula partly justifies why studying the expected squared magnitude of $\mathrm{d}_{\lambda}h_t$ in Section 2 makes general sense, as

$$\mathbb{E}\left[\left|\frac{\mathrm{d}h_{t,i}}{\mathrm{d}\theta}\right|^2\right] = |b_i|^2 S(\lambda_i, \bar{\lambda}_i). \tag{147}$$

**Magnitude-angle parametrization:** $\lambda = \nu(\omega_{\nu})\exp(i\theta(\omega_{\theta}))$. The calculations for this parametrization are similar to the previous one, with the following differences:

$$\frac{\mathrm{d}\lambda}{\mathrm{d}\omega_{\nu}} = \frac{\nu'(\omega_{\nu})\exp(i\theta(\omega_{\theta}))}{2} \tag{148}$$

$$\frac{\mathrm{d}\bar{\lambda}}{\mathrm{d}\omega_{\nu}} = \frac{\nu'(\omega_{\nu})\exp(-i\theta(\omega_{\theta}))}{2} \tag{149}$$

$$\frac{\mathrm{d}\lambda}{\mathrm{d}\omega_{\theta}} = \frac{i\nu(\omega_{\nu})\theta'(\omega_{\theta})\exp(i\theta(\omega_{\theta}))}{2} \tag{150}$$

$$\frac{\mathrm{d}\bar{\lambda}}{\mathrm{d}\omega_{\theta}} = -\frac{i\nu(\omega_{\nu})\theta'(\omega_{\theta})\exp(-i\theta(\omega_{\theta}))}{2}. \tag{151}$$

After some calculations we obtain

$$\frac{\mathrm{d}^2 L}{\mathrm{d}\omega_{\nu,i}\,\mathrm{d}\omega_{\nu,j}} = \frac{\nu'(\omega_{\nu,i})\nu'(\omega_{\nu,j})}{2}\mathrm{Re}[\mathrm{e}^{i(\theta(\omega_{\theta,i})+\theta(\omega_{\theta,j}))}A_{ij} + \mathrm{e}^{i(\theta(\omega_{\theta,i})-\theta(\omega_{\theta,j}))}B_{ij}] \tag{152}$$

$$\frac{\mathrm{d}^2 L}{\mathrm{d}\omega_{\nu,i}\,\mathrm{d}\omega_{\theta,j}} = \frac{\nu'(\omega_{\nu,i})\nu(\omega_{\nu,j})\theta'(\omega_{\theta,j})}{2}\mathrm{Im}[-\mathrm{e}^{i(\theta(\omega_{\theta,i})+\theta(\omega_{\theta,j}))}A_{ij} + \mathrm{e}^{i(\theta(\omega_{\theta,i})-\theta(\omega_{\theta,j}))}B_{ij}] \tag{153}$$

$$\frac{\mathrm{d}^2 L}{\mathrm{d}\omega_{\theta,i}\,\mathrm{d}\omega_{\nu,j}} = \frac{\nu(\omega_{\nu,i})\theta'(\omega_{\theta,i})\nu'(\omega_{\nu,j})}{2}\mathrm{Im}[-\mathrm{e}^{i(\theta(\omega_{\theta,i})-\theta(\omega_{\theta,j}))}A_{ij} + \mathrm{e}^{i(\theta(\omega_{\theta,i})-\theta(\omega_{\theta,j}))}B_{ij}] \tag{154}$$

$$\frac{\mathrm{d}^2 L}{\mathrm{d}\omega_{\theta,i}\,\mathrm{d}\omega_{\theta,j}} = \frac{\nu(\omega_{\theta,i})\theta'(\omega_{\theta,i})\nu(\omega_{\theta,j})\theta'(\omega_{\theta,j})}{2}\mathrm{Re}[-\mathrm{e}^{i(\theta(\omega_{\theta,i})+\theta(\omega_{\theta,j}))}A_{ij} + \mathrm{e}^{i(\theta(\omega_{\theta,i})-\theta(\omega_{\theta,j}))}B_{ij}]$$
$$\tag{155}$$

## C.3   Experimental details

We use the linear RNN architecture defined in Appendix A.1 as teacher and implement our experiments in JAX [63], using the default Flax [64] implementation of RNNs and the LRU implementation of Zucchet et al. [65]. Code is available here.

We initialize RNNs in the same way we initialized the teacher, and initialize the eigenvalues of the LRU and other complex-valued networks with magnitude in $[\nu_0, 1]$ and angle within $[-\theta_0, \theta_0]$.

Given that we are interested in the optimization properties of the different architectures, we only report training losses and do not perform any cross validation.

Here are additional details related to the different figures:

– **Figure 3**: see Tables 1 and 2.

– **Figure 4 and 9**: for panels A and B, we use $\nu_0 = 0.99$ and draw $A$ in a slightly different manner to the one described above (we directly draw the eigenvalues and eigenvectors so that we have two pairs of complex eigenvalues). We use automatic differentiation to compute the Hessian. For panels C and D, we use the same setup as described in Table 2, but keep the learning rate constant over the course of learning. We report the effective learning rate at the end of learning.

– **Figure 10**: for panels A, B and C, we draw the magnitude and angle of 10 $\lambda$ values independently, uniformly in $[\nu_0, \frac{1+\nu_0}{2}]$ and $[-\theta_0, \theta_0]$. Importantly, this means that there are no conjugate pairs, which leads to more diagonal Hessian matrices at optimality than in Figure 4. For panel D, see Table 3.

– **Figure 11**: same setup as for Figure 3.

As a rule of thumb, each LRU (or complex-valued diagonal network) experiment takes 3 minutes on a consumer-scale GPU (NVIDIA GeForce RTX 3070) and each RNN experiment takes 10 minutes on a CPU. The scans behind the results reported in the different figures require on the order of few hundreds run each. Including our preliminary explorations, the results we report in this section required 30 days of compute, one third of it on GPUs and two thirds on CPUs.

## C.4   Additional analyses

### C.4.1   Structure of the loss landscape for LRUs and S4

In the main text, we only provide an analysis of the loss landscape for the fully connected linear recurrent neural network and its complex-valued diagonal counterpart. We here complete this result by performing the same analysis for the LRU and S4. Given that S4 involves some form of parameter sharing between the magnitude and the angle of the recurrence complex eigenvalues through $\Delta$, which is required to avoid the explosion of the angle gradients, we are particularly interested in observing whether it fully mitigates or not the gradient explosion effect. The results of Figure 9.B and D do not reveal such benefits: the loss landscape does not have high curvature on the $\omega_A^{\mathrm{im}}$ direction, but it is moved in the $\omega_\Delta$ direction. We haven't investigated whether qualitative changes arise when changing the input distribution.

|  | RNN | LRU |
|---|---|---|
| Batch size | 128 | |
| Sequence length | 300 | |
| Hidden neurons (teacher) | 10 | |
| Input / output dimension | 1 | |
| $\nu_0$ | $\{0.32, 0.68, 0.84, 0.92, 0.96, 0.98, 0.99\}$ | |
| $\theta_0$ | $\pi$ | |
| Hidden neurons (student) | 64 | |
| log learning rate | $[-5, -4.5, -4, -3.5, -3, -2.5]$ | $[-2.5, -2, -1.5, -1, -0.5]$ |
| Optimizer (schedule) | Adam (cosine) | |
| Initialization | $[\nu_0 \text{ teacher}, \nu_0 = 0]$ | $\nu_0$ teacher |
| Number iterations | 10k | |
| Seeds | 10 | |

Table 1: **Experimental details for Figure 3.A.** We use $[\cdots]$ to denote hyperparameters that were scanned over with grid search and $\{\cdots\}$ to denote the variables of interest for the figure. We chose the learning rates for the two architectures on preliminary scans and verified that non of the extreme learning rates were optimal in the final scan. For the RNN, we found that initializing with $\nu_0 = 0$ gave better results than initializing with the same distribution the teacher has, so we included this choice in the scan.

|  | RNN / BLOCK DIAG. RNN | COMPLEX DIAG. RNN / LRU |
|---|---|---|
| Batch size | 128 | |
| Sequence length | 300 | |
| Hidden neurons (teacher) | 10 | |
| Input / output dimension | 1 | |
| $\nu$ | 0.99 | |
| $\theta_0$ | $\pi$ | |
| Hidden neurons (student) | 64 / 64 and 128 | 64 |
| log learning rate | $[-5, -4.5, -4, -3.5, -3, -2.5]$ | $[-2.5, -2, -1.5, -1, -0.5]$ |
| Optimizer (schedule) | Adam (cosine) | |
| Initialization | $[\nu_0 \text{ teacher}, \nu_0 = 0]$ | $\nu_0$ teacher |
| Number iterations | 10k | |
| Seeds | 10 | |

Table 2: **Experimental details for Figure 3.B.** We use $[\cdots]$ to denote hyperparameters that were scanned over with grid search and $\{\cdots\}$ to denote the variables of interest for the figure. We chose the learning rates for the two architecture types on preliminary scans and verified that non of the extreme learning rates were optimal in the final scan. For the RNN, we found that initializing with $\nu_0 = 0$ gave better results than initializing with the same distribution the teacher has, so we included this choice in the scan. For the RNNs, we used 64 neurons for the "RNN" entry, 64 for the "block diagonal" one, and 128 for the "more neurons" one.

|  | RNN | COMPLEX DIAG. RNN / LRU |
|---|---|---|
| Batch size | 128 | |
| Sequence length | 300 | |
| Hidden neurons (teacher) | 10 | |
| Input / output dimension | 1 | |
| $\nu_0$ | 0.99 | |
| $\log(\theta_0/\pi)$ | $\{-2, -1.5, -1, -0.5, 0\}$ | |
| Hidden neurons (student) | 64 | |
| log learning rate | $[-4.5, -4, -3.5, -3]$ | $[-3.5, -3, \cdots, -0.5, 0]$ |
| Optimizer (schedule) | Adam (cosine) | |
| Initialization | $[\nu_0 \text{ teacher}, \nu_0 = 0] + \theta_0$ teacher | $\nu_0$ teacher $+ \theta_0$ teacher |
| Number iterations | 10k | |
| Seeds | 10 | |

Table 3: **Experimental details for Figure 10.** We use $[\cdots]$ to denote hyperparameters that were scanned over with grid search and $\{\cdots\}$ to denote the variables of interest for the figure. We chose the learning rates for the two architectures on preliminary scans and verified that non of the extreme learning rates were optimal in the final scan. For the RNN, we found that initializing with $\nu_0 = 0$ gave better results than initializing with the same distribution the teacher has, so we included this choice in the scan.

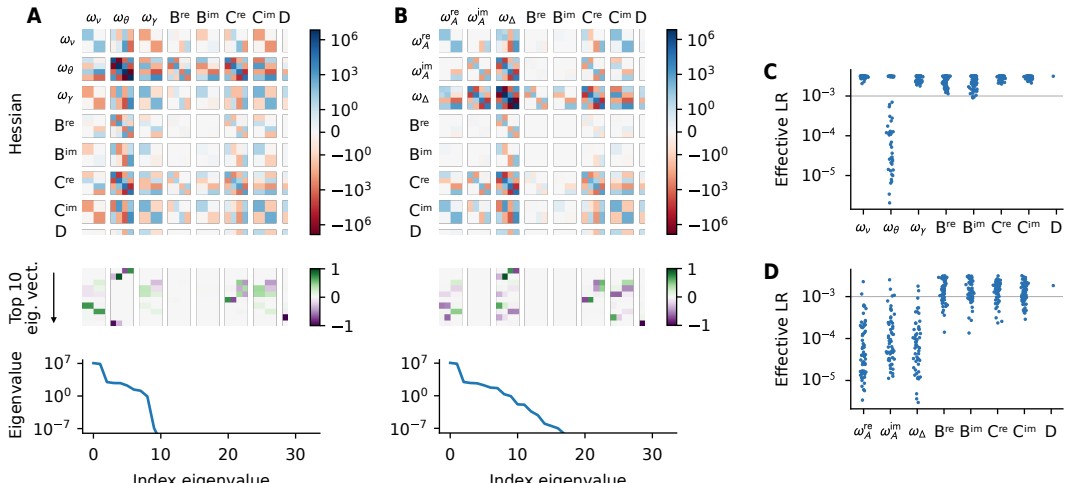

Figure 9: **Equivalent of Figure 4 for the LRU (A, C) and S4 (B, D).** In the LRU, the exponential parametrization of the magnitude $\nu = \exp(-\exp(\omega_\nu))$ efficiently mitigates the Hessian explosion but not the one of the angle $\theta = \exp(\omega_\theta)$, consistently with the theoretical and empirical evidence we have accumulated so far. In B, $\Delta$ is set to 0.01 in the S4 architecture. Setting it to 1 (not plotted here) leads to an Hessian at optimality that has large entries on all recurrent parameters $\omega_A^{\text{re}}$, $\omega_A^{\text{im}}$ and $\omega_\Delta$, similarly to the behavior we observed for the complex diagonal RNN studied in the main text. For panel D, we initialized $\Delta$ at 1 given that this is the initialization that yielded best performance. We note that we didn't find qualitative changes in this plot when changing the initialization scheme of S4.

### C.4.2 Concentrating eigenvalue distributions

The goal of this experiment is to better understand how the concentration of eigenvalues $\lambda$ affect the learning dynamics. For fully connected RNNs, there is no reason to expect a major change in behavior. However, it is different for diagonal RNNs. The theoretical analysis we have done in Section C.2 provides us with the following insights. When the elements in the input sequence are uncorrelated, as it is the case here, the entries in the Hessian corresponding to two different eigenvalues increase if they are aligned or conjugate to each other, and if their magnitude is large. We therefore expect that, as the interval on which the angle of the teacher's eigenvalues shrinks ($\theta_0 \to 0$),

those eigenvalues will be more likely to be "similar" to each other. This results in large non-diagonal terms, as we confirm in Figure 10.A, B and C. The LRU suffers less from this problem thanks to its reparametrization, which reduces the overall magnitude of Hessian entries related to the magnitude, and partly the one of angle parameters (when it is a small positive number). As a consequence, the performance between these two architectures increases as $\theta_0 \to 0$, as seen on Figure 10.D.

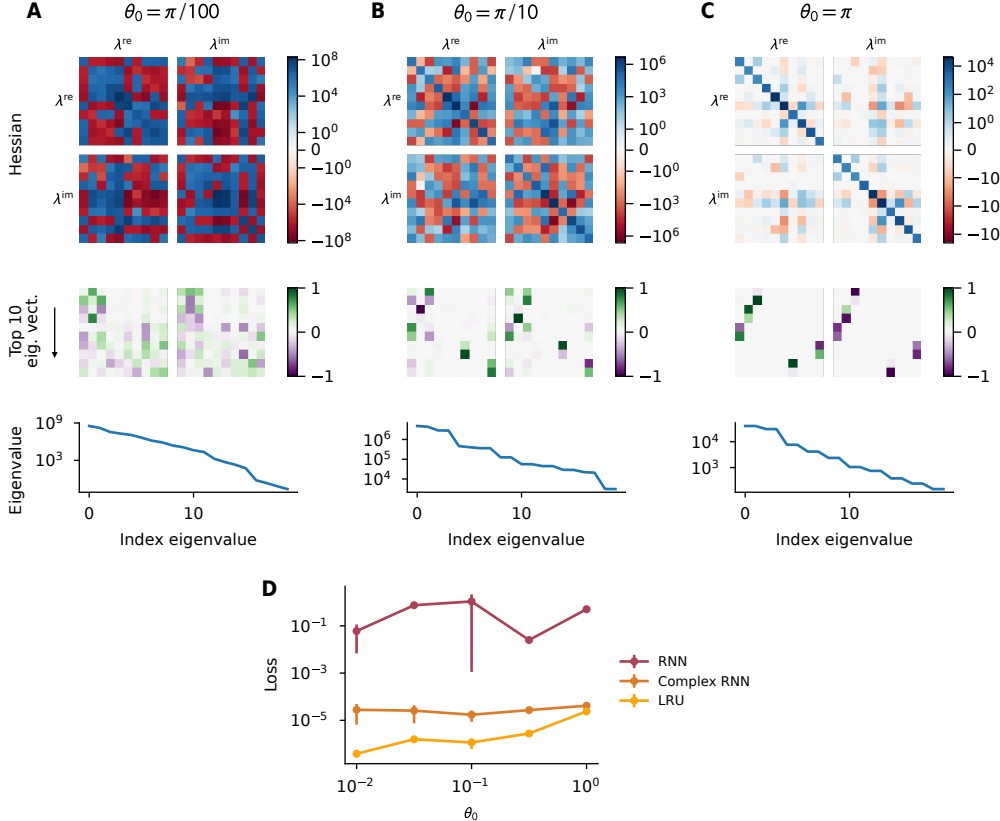

Figure 10: **Concentrating eigenvalues make the Hessian less diagonal ($\theta_0 \to 0$) and consequently increases the gap between the LRU and the complex-valued diagonal RNN. A, B, C.** Hessian of the loss with respect to the $\lambda$ parameters in the complex-valued diagonal RNN. The Hessian is computed through the theoretical formula of Equation 143; computing it numerically marginally affects the results. Consistently with the intuition we developed in Section C.2, concentrating the eigenvalues affect the structure of the loss landscape. It makes the Hessian at optimality less diagonal and Adam cannot efficiently compensate it. The LRU does not suffer as much from this problem, and the gap between the two architecture widens as $\theta_0 \to 0$.

### C.4.3 Impact of the number of heads in fully connected linear recurrent networks

In Figure 3, we have shown that constraining the connectivity matrix to be block diagonal with blocks of size $2 \times 2$ lead to a critical boost in performance. Further analysis revealed that this arises as the Hessian becomes more diagonal and Adam can thus better compensate for gradients explosion. Here, we study this behavior in more detail by interpolating between the fully connected case and the block diagonal one. This can be achieved by increasing the number of independent heads from 1 (fully connected case) to 32 ($2 \times 2$ block-diagonal connectivity matrix, as we have 64 hidden neurons). In particular, we are interested in understanding how big heads can be while keeping this performance boost. We plot the final performance, as well as the evolution of the effective learning rate for the $A$ matrix over the course of learning on Figure 11. We find that slightly bigger heads, until $4 \times 4$ (which corresponds to 16 heads), yield similar benefits. Additionally, the learning rate analysis reveals that the adaptive learning rates of Adam can more selectively compensate for potential gradient explosion cases as the number of heads increases, allowing for bigger learning rates overall and better performance.

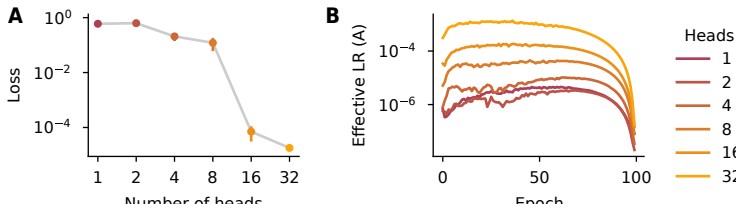

Figure 11: Evolution of the performance (**A**) and effective learning rates for the $A$ connectivity matrix (**B**) of a linear recurrent neural network as we vary the number of heads, keeping the overall number of hidden neurons fixed. It should be noted that increasing the number of heads decrease the total number of parameters, as the matrix $A$ gets sparser.

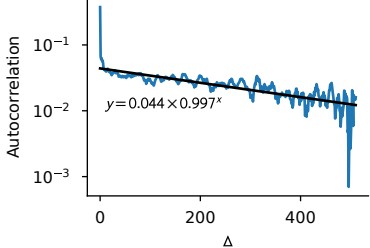

Figure 12: **The empirical autocorrelation function (averaged over feature dimensions) of the BERT embeddings used in Section 5 can be approximated as a sum of two exponentially decaying functions.** The blue line represents the autocorrelation function $R_x^{\text{empirical}}(\Delta)$ of the BERT embeddings of the Wikipedia dataset. As a first approximation, it is equal to $R_x^{\text{empirical}}(\Delta) \approx 0.376\delta_{\Delta=0}$. For a more refined approximation, we perform a linear regression of the log autocorrelation against $\Delta$, shown by the black line. This yields the following approximation: $R_x^{\text{empirical}}(\Delta) \approx 0.332\delta_{\Delta=0} + 0.044 \times 0.997^{\Delta}$.

## D    Signal propagation in randomly initialized deep recurrent neural networks

### D.1    Experimental setup

We detail the experimental setup used in Section 5. We select the first 512 tokens from 1024 random sequences in the Wikipedia dataset [66] and pass them through the BERT [67] tokenizer and embedding layer. This yields a dataset of 1024 examples with length 512 and feature dimension 724. Figure 12 shows the autocorrelation function of these inputs, revealing that the i.i.d. assumption serves ($\rho = 0$) as a good first order approximation. This validates the relevance of our toy experiments for studying signal propagation in more realistic settings. To refine this approximation, we can include a high correlation term ($\rho$ close to 1).

We examine realistic networks comprising 4 blocks with the following structure: a recurrent layer, a non-linearity, a gated linear unit [58, GLU] and a skip connection. By default, we omit normalization layers, but when included, as in Figure 5.C, we place one normalization layer before the recurrent layer and another one before the GLU. All the layers involved contain 256 neurons. We also incorporate a linear encoder at the beginning of the network and a linear decoder at its end.

The loss that we use is a next-token mean-squared error, defined as

$$L_t = \frac{1}{2}\|\hat{x}_t(x_{1:t-1}) - x_t\|^2 \tag{156}$$

where $\hat{x}_t(x_{1:t-1})$ represents the prediction of the network. Figure 5 reports the average squared value of the hidden state or the gradient. This average is computed over sequences, but also over all neurons / parameters and over all time steps. We compute gradients using batches of size 8.

In Figure 5 we vary $\nu_0$, which controls the magnitude of the eigenvalues of the recurrent Jacobian. Specifically, we sample those magnitudes in the interval $[\nu_0, (1 + \nu_0)/2]$. For the complex-valued diagonal RNN and the LRU, we apply the LRU initialization. For the LSTM, we use the Chrono

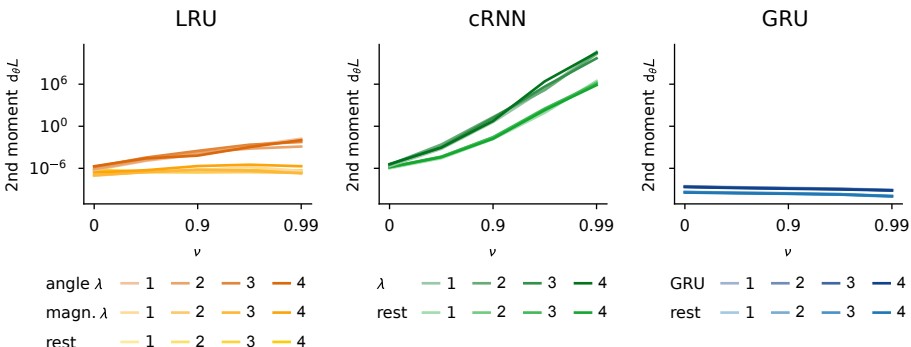

Figure 13: **Gradient magnitudes are independent of the layer.** This figure presents similar plots to Figure 5.B, except that parameters from different layers are no longer aggregated together. Instead, each parameter group of each layer has its own line. The indices in the legend correspond to layer indices.

initialization proposed by Tallec and Ollivier [29]: it initializes the forget gate biases such that, when the input $x$ and the hidden state $h$ are equal to 0, the time constant associated to $f$ is uniformly sampled from $\left[\frac{1}{1-\nu_0}, \frac{2}{1-\nu_0}\right]$ and the input gate $i$ is equal to $1 - f$.

While Figure 5.B presents aggregated gradients over layers, Figure 13 offers a layer-wise version of this analysis. It reveals that the layer index does not significantly impact gradient magnitude. Surprisingly, given that the hidden states of the complex RNN gets larger with depth (c.f. Figure 5.A), this result might seem unexpected for cRNNs. We can attribute this to the backpropagated error signals also being amplified during the backward pass, as discussed in Section 2.2. In the first layers, hidden states are small and errors are large, while in the in last layers, errors are small and hidden states are large. Consequently, the gradient magnitude remains relatively constant accross layers. For the GRU, the gradient magnitude reported in Figure 5 for the non-GRU parameters included the linear encoder and decoder. As the encoder gradients are the dominating ones, this explains why the gradient magnitude for the non-GRU parameters is smaller in the layer-wise analysis.

### D.2 Can gated RNNs be considered diagonal?

In Section 3.2, we argued that the diagonal linear setting we focused our theory on can be a decent proxy for more general gated RNNs, whose $\lambda$ values can depend on both the inputs $x$ and the hidden state $h$. Here, we assess whether this holds true at initialization. To that end, we study how the Jacobian $\frac{\mathrm{d}h_{t+\Delta}}{\mathrm{d}h_t}$ of a GRU evolves as $\Delta$ grows. For the linear diagonal regime to be a good proxy, two necessary conditions must be met: First, all the non-diagonal terms should be negligible compared to the diagonal ones. Second, the diagonal terms should decay similarly as if their $\lambda$ values were independent of $x$ and $h$.

In Figure 14, we report the evolution of all the diagonal terms of this Jacobian and a random subset of the non-diagonal ones. Our findigns indicate that the non-diagonal terms are much smaller than the diagonal ones under the standard initialization, supporting the first necessary condition. Additionally, input and hidden state-dependent gates do not qualitatively change the decay of the diagonal terms, particularly when the network is initialized with $\lambda$ values close to 1 (which correspond to large time constants). Furthermore, we find that increasing the strength of all the hidden state to gate connections ($W_{hn}, W_{hr}, W_{hz}$) breaks the diagonal-like behavior and eventually leads to exploding Jacobians. However, this only occurs at values that are much larger (3 times more in this case) than the default initialization of these weights (orthogonal initialization).

In conclusion, those results confirm that the theoretical setting we have considered in this paper is a good proxy for studying signal propagation in realistic recurrent networks. While we have focused our analysis on GRUs, we expect these results to hold for other architectures such as LSTMs, Mamba, or Hawk. For the last two, given that the different gates only depend on the input, we expect the matching with our theory to be even stronger (c.f. Figure 14.C $\sigma_h = 0$, which captures this regime).

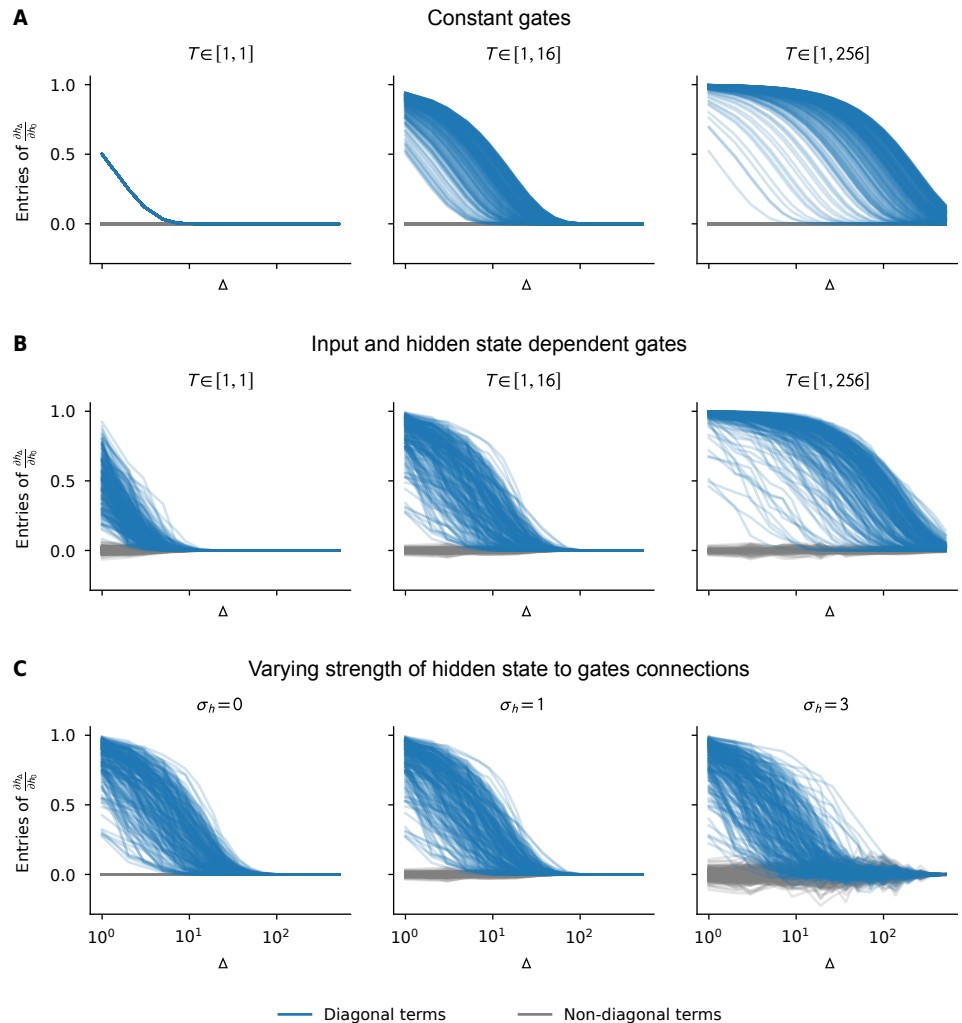

Figure 14: **GRUs behave like diagonal linear networks.** This figure illustrates the evolution of the recurrent Jacobian $\frac{\mathrm{d}h_\Delta}{\mathrm{d}h_0}$ of a GRU, when provided with the BERT embeddings of a sentence extracted from the Wikipedia dataset. **A.** In the first row, we take the forget gates to be independent of $x$ and $h$ and we sample them with the Chrono initialization [29] for different intervals. The resulting network is linear and diagonal, similar to what we have studied in the theory. These plots therefore serve as visual reference for comparison with the realistic case. **B.** The second row shows the same plots as the previous row, except that the gates are now dependent on the inputs $x$ and on the hidden states $h$. As mentioned in A.4, we initialize all the linear layers taking $x$ as input with LeCun initialization and the layers taking $h$ as input with orthogonal initialization. The recurrent Jacobian evolves similarly than in the constant case, particularly on slowly decaying dimensions (large $T$ values). **C.** In the row, we aim to break the diagonality of the model by increasing the strength $\sigma_h$ of the hidden state to gate connections, for $T \in [1, 16]$. The case $\sigma_h = 0$ corresponds to gates that only depend on $x$, similar to architectures like Mamba or Hawk. The plot with $\sigma_h = 1$ is the same as the middle one in B. For $\sigma_h = 3$ and higher, the diagonality progressively breaks and the recurrent Jacobian eventually explodes. We note that these plots were obtained from a single example. Yet, we have found the behavior we report here to be typical of the general behavior.

### D.3 Does our theory apply to gated RNNs?

Having established that gated RNNs behave similarly to the linear diagonal RNNs considered in our theoretical investigation, a question arises: How well can our theory describe signal propagation in gated RNNs on realistic data? To address this, we study a simplified version of the GRU network (similar to the one studied by Chen et al. [51]), which incorporates a realistic gating mechanism:

$$f_{t+1} = \sigma(W_{fx}x_{t+1} + W_{fh}h_t + b_f) \tag{157}$$
$$h_{t+1} = f_{t+1} \odot h_t + (1 - f_{t+1}) \odot x_{t+1}. \tag{158}$$

As in the rest of this section, we provide BERT embeddings of sentences from the Wikipedia dataset as inputs.

To apply our theory to this architecture, we must address two main challenges:

1. The gate $f_{t+1}$ depends on both $h_{t+1}$ and $x_t$, making it non-constant. Based on our empirical results from the previous section, we reasonably approximate $f_{t+1} \approx \sigma(b_f) =: \lambda$, ignoring this dependency.

2. Our theoretical derivations have overlooked cases where the recurrence strength $\lambda$ normalizes the input $x_t$. When considered, we detached the normalization factor from the computational graph (as in Section B.3.2). However, we can extend our calculations from Section B.2 to accommodate this setting:

$$f(\alpha, \beta) := \frac{(1-\alpha)(1-\beta)}{1-\alpha\beta} \left( R_x(0) + \sum_{\Delta \geq 1} (\alpha^\Delta + \beta^\Delta) R_x(\Delta) \right) \tag{159}$$

$$\mathbb{E}[h_t^2] = f(\lambda, \lambda) \tag{160}$$

$$\mathbb{E}\left[ \left( \frac{\partial h_t}{\partial \lambda} \right)^2 \right] = \left. \frac{\partial^2 f(\alpha, \beta)}{\partial \alpha \partial \beta} \right|_{\alpha=\lambda, \beta=\lambda}. \tag{161}$$

Note that we obtain $\mathbb{E}[(\frac{\partial^2 h_t}{\partial b_f^2})^2]$ by multiplying $\mathbb{E}[(\frac{\partial h_t}{\partial \lambda})^2]$ by $\sigma'(b_f)^2$.

With these adjustments in place, we can now empirically test our theoretical predictions. For simplicity, we approximate the auto-correlation function $R_x$ (blue line in Figure 12) as $R_x(\Delta) \approx 0.332\delta_{\Delta=0} + 0.044 \times 0.997^\Delta$. Figure 15 presents our results, which reveal:

- An almost perfect match between theory and practice for constant gates, confirming that our sample size is large enough.

- A very precise, though not perfect, match for context-dependent gates.

- The variance of $h_t$ and $\frac{\partial h_t}{\partial b_f}$ shows minimal dependence on $\lambda$, indicating that the magnitude of error signals received by $b_f$, $W_{fh}$, and $W_{fx}$ are largely independent of the time constants encoded in the network.

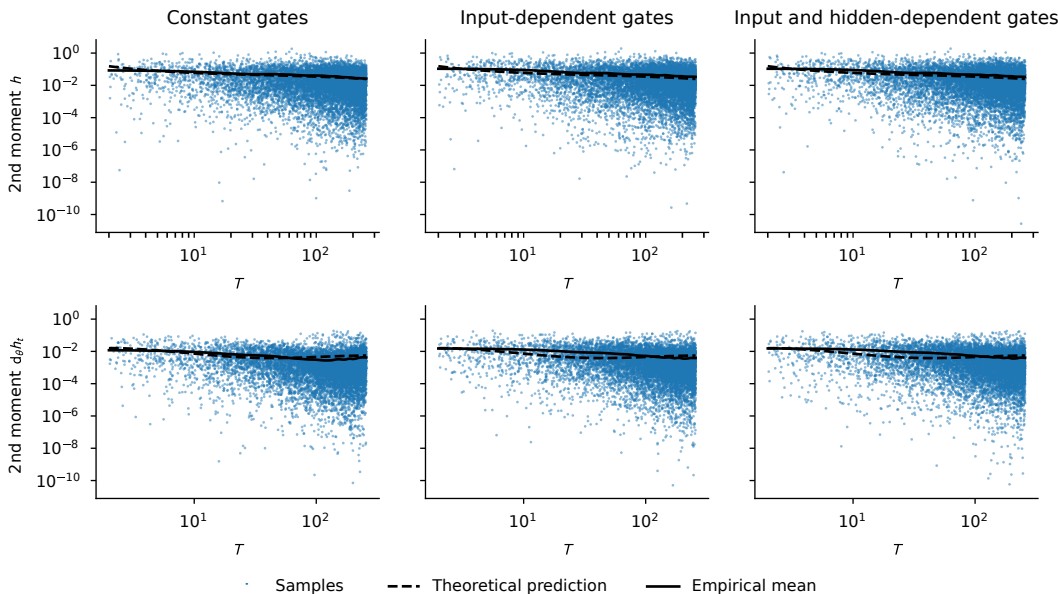

Figure 15: **The theory developed for linear diagonal recurrent networks captures signal propagation within gated recurrent neural networks.** The different samples were obtained as follows: $100$ different randomly initialized networks are given a different input sequence of length $512$. The biases of the forget gates $b_f$ are initialized with Chrono initialization for $T \in [1, 256]$. For each of these models / sequences, we measure $h_{512,i}^2$ and $\mathrm{d}_{b_{f,i}} h_{512,i}^2$ ($i$ being the index of one of the $256$ hidden neurons). We report this measurement as a function of the time constant $T$ encoded by the neuron ($\lambda = T/1 + T$). The empirical mean is obtained with a kernel regression with the Gaussian kernel $\mathcal{K}(a, b) := \exp(-(a - b)^2/100)$. The theoretical prediction comes from the approach described in Section D.3.

