# OpenReview forum: "Recurrent neural networks: vanishing and exploding gradients are not the end of the story"
_NeurIPS.cc/2024/Conference — NeurIPS 2024 poster_

### Official Review · Reviewer_xzKN · 2024-06-16

**Soundness:** 2
**Presentation:** 2
**Contribution:** 2
**Rating:** 5
**Confidence:** 4

**Summary:**

This paper studies the vanishing and exploding gradient phenomenon in RNNs. In particular, it answer the question: "is solving those issues really enough to ensure well-behaved loss landscapes?" This paper reveals the importance of the element-wise recurrence design pattern combined with careful parametrizations in mitigating the scale issue of hidden states.

**Strengths:**

1. It is good to see people discuss the complex eigenvalue setting as previous work seems to be limited to real eigenvalues (such as HiPPO)
2. The teacher-student analysis is well-presented and clarifies the understanding of complex systems within this context.

**Weaknesses:**

1. The assumption of inputs being wide-sense stationarity seems strong. Does it include the setting such as the Cifar10 dataset?
2. The argument of input normalization seems hand-waving.
	1. When $\lambda \to 1$, the forward pass and backward pass are exploding with a different speed: Forward is asymptotically $1/(1-\lambda)$ while backward is (at least) asymptotically $1/(1-\lambda)^2$.
	2. It should be proper to say normalization can relax one scale issue but it cannot achieve the scaling both at the same time.
3. The paragraph "On the Importance of Adaptive Learning Rates" addresses an essential topic but lacks direct training curves.
	1. While the local Hessian analysis provided is valuable, it doesn’t assure improved global performance.
	2. It would be more effective and indeed necessary to include training curves comparing results using SGD and adaptive learning rates (such as AdamW). This would clearly illustrate the advantages and need for adaptive optimizers.
4. Related works on parameterizations and orthogonal matrix is limited:
	1. https://proceedings.mlr.press/v80/zhang18g.html
	2. https://arxiv.org/abs/1909.09501: This work proposes a parameterised approach to train orthogonal matrix.
	3. https://arxiv.org/abs/1901.08428
	4. https://openreview.net/pdf?id=ryxepo0cFX
	5. https://arxiv.org/abs/2311.14495: This work proposes the relationship between parameterization and gradient in RNNs/SSMs.
	6. https://arxiv.org/pdf/2006.12070
5. I am willing to increase the score if the above issues are addressed.

**Questions:**

1. On Page 4, Equation 5 focuses on the second-order expectation of $h_t$​. It would be helpful to understand the specific reason for choosing a second-order expectation over a first-order one. Generally, implementations do not involve the second moment of hidden states. Why is the first-order expectation not considered in this case? Does first-order suffer from the same problem?
	1. A similar forward-pass blow-up has been proved in the LRU(Orvieto, 2023) paper, proposition 3.3. What is the difference in terms of assumption and technique between your result and the LRU result?
2. On Page 5, in the paragraph titled "What About Complex Numbers?", you discuss the concept of polar parametrization. This raises an important question: do all parametrizations encounter the same issues as the polar parametrization? If they do, demonstrating this could effectively highlight the limitations of using complex numbers in a polar setting. If not, it may be worthwhile to find the good parameterization.  Currently, the exploration of polar parametrization seems quite restricted.
3. Your title is "vanishing and exploding gradients are not the end of the story". A question based on the saying in Page 9 Figure 5 Panel C is:
	1. Does layer normalization and batch normalization resolves the all problems?
	2. If not, I think it's hard to claim "Layer normalization keeps the overall gradient magnitude under control".

**Limitations:**

1. This paper investigates how the hidden states of recurrent models contribute to learning long-term memory. There is very little research available to explain why the mamba model, which uses both gating and state-space models, is successful.
	1. The discussion on Gated RNNs in lines 205-220 lacks rigorous substantiation, relying primarily on speculative assertions to illustrate the impact of gating mechanisms.
	2. The paper does not clearly establish whether the proposed methodology is effective within a gated framework.

---

> ### Author Rebuttal · Authors · 2024-08-06
>
> We thank the reviewer for their thorough feedback. We address their concerns below.
>
> **Weaknesses**
>
> 1. **Wide-sense stationarity.** A discussion of this assumption was indeed missing. We address it in the global response. In short, this assumption is standard in the analysis of linear time filters (see e.g. the Proakis and Manolakis book) and is mostly for convenience in making calculations easier (see e.g. the classical Wiener–Khinchin theorem). When it is violated, our theory does not perfectly predict signal propagation, but our results hold qualitatively and almost quantitatively (c.f. Fig 1 and 3 of the PDF, which test our predictions in a practically relevant setting). This assumption is not a significant limitation of our theory, as using the empirical autocorrelation function leads to very similar results (see Fig 3 of the PDF, left col).
> 2. **Input normalization.** The reviewer accurately points out that the forward and backward passes (for the $\lambda$ parameters, but not for the inputs $x$) explode at different speeds and that normalization alone cannot solve it. Normalization is a way to ensure that signals, both hidden states and errors, stay constant over the network's depth. Therefore, its role is to keep the magnitude of the hidden states bounded. However, it does not entirely avoid the explosion of the recurrent parameter gradients, as hinted by the reviewer. This is where reparametrization is required. We hope that this clarifies our argument. If not, can the reviewer point out the source of confusion in this argument?
> 3. **Adaptive learning rates.** The Hessian analysis indeed only captures the geometry of the loss landscape around optimality. Outside these regions, this only corresponds to part of the Hessian and includes some approximation. However, we would like to point out that this approximation is very standard, e.g., [Sagun et al. 2018](https://arxiv.org/pdf/1706.04454), and we find that the insights we gained by studying the Hessian at optimality hold for the rest of training.
> Regarding the training curves of SGD/Adam: This section was indeed missing some discussion of SGD. We found that it is almost impossible to train such recurrent networks with SGD. The Hessian at optimality gives us a hint as to why this happens: the Hessian on the recurrent parameters (e.g., $\lambda$) is much larger than the one on feedforward parameters (e.g., $D$). As a result, to keep SGD stable, we need tiny learning rates, and the feedforward parameters barely change. The training curves for SGD are therefore almost flat. On the contrary, Adam uses adaptive LRs for different parameters, thereby avoiding this problem. We will mention this point and include some training curves in the next version of the paper.
> 4. **References**. We thank the reviewer for these references.
>
> **Questions**
>
> 1. **Why second-moment?** We focus on the second moment for multiple reasons. First, the first moment would be 0 as long as the input has a mean of 0, which is the traditional regime in deep learning. Second, this enables us to partially capture the norm of the different quantities at hand. Third, this quantity appears, up to some kind of data-dependent change of norm, in the Hessian, see e.g. Eq. 11. Finally, we would like to note that studying second-order quantities is standard in deep learning, a classical example is the seminal Glorot & Bengio initialization paper (2010).
> Regarding the comparison with Prop 3.3 of the LRU: our Eq. 4 is indeed very close to their result. Here, we generalize it to more general input distributions.
> 2. **On complex numbers.** In our paper, we discuss both polar and real-imaginary parametrizations, which are arguably the most usual ones, and show that they suffer from explosion behaviors. We tried our best to find an "optimal" parametrization for the polar case, given that we know how to control the magnitude of the gradient w.r.t. $|\lambda|$. However, we found that the most natural choices all have some issues (cf. A.3.2). It does not mean that it is impossible to find a better parametrization, just that we could not. We are happy to consider any suggestions that the reviewer may have.
> 3. **On normalization.** In Fig 5, we show that overall gradient magnitudes remain bounded with layer normalization, hence “the overall gradient magnitude [is] under control”. However, this is not a proper solution as different hidden neurons share the same normalizers. It follows that a neuron with a high $\lambda$ will heavily influence the normalization factor and thus neurons with smaller $\lambda$ values will be ignored. We would also like to highlight that the LRU paper already includes a similar argument.
>
> **Limitations**
>
> First, we would like to point out how rich the learning dynamics of the simple time-invariant recurrent network are and that analyzing it in depth is not a trivial task. We are convinced that this analysis alone is already a strong contribution to the field. That said, the reviewer’s remark prompted us to analyze more carefully the gated case on our task of Sec 5. We study a GRU as it is the simplest gated architecture. We report those results in the global answer. Overall, we find that diagonality almost holds under standard initialization, for the reasons we mention in lines 205-220, and that our theory captures signal propagation in this regime. However, it is likely that, during training, the architecture becomes less linear and that our theory cannot capture signal propagation anymore. Such architectures have been studied using the mean field framework by Chen et al. 2018 [50]. Their result captures non-linearity but is arguably harder to interpret. We believe that our work provides a complementary and more intuitive understanding of signal propagation while being less general.
>
> We hope that our rebuttal convinces the reviewer of the relevance and soundness of our work and that they may consider reevaluating our paper.

---

> ### Comment · Reviewer_xzKN · 2024-08-13
>
> I have carefully reviewed both the reviews and the rebuttals.
>
> However, I believe that the section on the wide-sense stationary assumption may not be essential for deriving the key results. Estimating hidden states could potentially be approached deterministically with bounded inputs, which might more effectively highlight the advantages of specific parameterizations, initializations, or normalizations.
>
> While the inclusion of the stationary property facilitates a discussion of the second-order moment, this approach seems somewhat forced. The bound on the second-order moment within a random process context does not provide a significantly more valuable insight. Although generalizing to more diverse input distributions is a noteworthy contribution, it is crucial to clearly define the boundaries of this generalization. Without specifying the limits of input expansion, the contribution may not be fully strengthened. It remains unclear how this paper’s contribution stands apart from the LRU paper and what kinds of stronger understanding can be derived for complex model such as gated.
>
> Therefore, I will increase my score from 4 to 5 and encourage the authors to address the issues related to the stationary assumption and to clarify the improvements over the LRU paper in the next version.

---

> > ### Author Response · Authors · 2024-08-13
> >
> > We thank the reviewer for updating their score.
> >
> > The bounded input assumption the reviewer is suggesting is an interesting one that we did not consider. Such an approach gives an upper bound on the quantities we study that would correspond to the case $\rho=1$ in our analysis (up to a multiplicative constant of course). Our approach has two main benefits compared to this one:
> > - It also provides a lower bound on the second-moments considered given that it is an equality
> > - Our upper bound is more precise when the auto correlation function decreases quickly (i.e. $\rho < 1$, c.f. Fig 1.A), which is the more realistic case (see Figure 1 in our rebuttal PDF for an example)
> >
> > To get those results, we have to assume stationarity (which is arguably stronger than boundedness), which indeed does not hold in practice. However, given that we empirically found that our results almost perfectly hold when stationarity is not met, we believe that the benefits of this assumption largely outweigh the fact that it will never be exactly achieved in practice. We will include this discussion to the appendix in the next version of the paper. We hope that this argument convinces the reviewer of the interest of the stationarity assumption.

---

### Official Review · Reviewer_Bq7Y · 2024-07-10

**Soundness:** 3
**Presentation:** 3
**Contribution:** 3
**Rating:** 5
**Confidence:** 5

**Summary:**

In this work, the authors present an analysis of Recurrent Neural Networks (RNNs), focusing on problems which impede optimization as the length of input sequences and model memory increase. The authors first focus on the well-studied problem of vanishing/exploding gradients, arguing that this problem by itself does not explain the difficulties in optimization. The authors turn to the less studied "curse of memory", and argue that this issue persists for RNNs even when gradients are controlled to not explode or vanish. Finally, the authors study how some of the alternative recurrent architectures such as LSTMs and SSMs mitigate the curse of memory, and provide some suggestions for how optimization can be improved.

**Strengths:**

I am very intrigued by the approach taken in this work.

The analysis seems novel and the arguments provided are given in a straightforward and convincing manner. The theoretical and empirical results in the main body of the work all seem sound, and are augmented by substantial additional experiments in the appendices.

The authors' analysis results in clear explanations both for how current architectures mitigate the issues studied in this work, and the analysis suggests some directions that may lead to new architectural innovations in the future.

Despite many technical issues in the writing, the argument is presented in a clear way and the flow of the paper makes sense. See below.

**Weaknesses:**

I have three major issues with the paper in its current form. If issue 1 is addressed, I am willing to increase the score of the presentation from fair to excellent, which would bring the paper to a borderline reject. Issues 2 and 3 are more relevant to the soundness of the work, and I've lowered the score to a 2. If the authors address these, I will consider raising the score; however, I can't go above a 3 without a more substantial discussion of limitations, which I think might not be feasible within the rebuttal period; however, I am still open to adjusting this score depending on the discussion.

1. One of the more glaring issues with this work is with the writing. The authors need to take some time to carefully check for typos, incomplete sentences, and correct grammar. I have listed a few particular issues here:
  * line 30: remains -> remain
  * line 69: You say here that you are showing the gradients can explode, but in the next section say you are controlling the gradients so that they do not explode (lines 83-86). Am I missing something conceptually here that makes this not an inconsistency?
   * line 70: remains -> remain
   * Figure 1: the last sentence of the caption is unclear and seems to be missing some words.
   * line 86: should be "studying this behavior quantitatively", not the other way around
   * line 192: alleviates -> alleviate
   * line 283: characteristic -> are characteristic

if these issues are all fixed and the authors will take some time to carefully edit, I will update my score to reflect this effort. Overall the typos only affect clarity in a few spots.

2. In section 2.2 - you justify assumptions A) and B), but I couldn't find any justification of assumption C). Unless I am missing something, this needs to be provided, as assuming Wide-sense stationarity is not necessarily standard. At the very least, this assumption could be acknowledged as a limitation and room for future work.

3. There is no real discussion section at all, and there is particularly no discussion of limitations.

**Questions:**

See above - why is the assumption of Wide sense stationarity needed, and how is your analysis affected by not adopting the assumption?

**Limitations:**

No limitations are presented or discussed in the paper as far as I could find.

I see no ethical issues with this work.

---

> ### Author Rebuttal · Authors · 2024-08-06
>
> We appreciate the reviewer’s positive feedback on our paper and address their concerns below.
>
> 1. **Typos**. We thank the reviewer for their detailed report of our typos. It will help us make the manuscript as clear as possible and without typos. Even though the system does not allow for the upload of a revised version of the paper at this stage, we already implemented all the corrections in our latex file and made several additional passes to make sure everything is in the best shape. That said, we are happy to provide clarifications for the typos that affect the understanding of our argument.
> Regarding line 69: we mean that the gradients’ magnitude can explode when the memory increases (i.e., $\lambda \rightarrow 1$), even when we are outside the traditional exploding gradient regime (i.e., we enforce $\lambda \leq 1$). It should be noted that in the first case the gradients explode as the memory increases, whereas in the second case, they explode as the interval considered increases.
> Regarding Figure 1: “as” is missing (”The loss becomes sharper as information is kept…”).
> 2. **Wide-sense stationarity assumption.** A discussion of this assumption was indeed missing in the paper. We address it in the global response. In short, this assumption is standard in the analysis of linear time filters (see e.g. the Proakis and Manolakis book) and is mostly for convenience in making calculations easier (see e.g. the classical Wiener–Khinchin theorem). When it is violated, our theory does not perfectly predict signal propagation, but our results hold qualitatively and almost quantitatively (c.f. Figures 1 and 3 in the rebuttal PDF, which test our theoretical predictions in a practically relevant setting). Overall, this assumption is not a significant limitation of our theory, and using the empirical autocorrelation function leads to very similar results (see Figure 3 of the rebuttal PDF, left column).
> 3. **Discussion of the limitations.** As we discussed in the global response, the limitations of our work are mostly due to the simplicity of our model and the nature of the approach we are following.
> Regarding the model we study: While our results only apply to the non-gated case, we show that they still capture the gated case relatively accurately, despite the dynamics being significantly nonlinear (c.f. Figure 3 in the rebuttal PDF). However, there is little hope that our analysis extends to nonlinear vanilla RNNs.
> Regarding what we can achieve with such an approach: Studying signal propagation has its limits regarding the insights it can bring. We can mainly study the loss landscape through such an approach, and cannot say anything about, e.g., the memory capabilities of a given architecture.
>
> The reviewer's thoughtful comments have helped us clarify key aspects of our work. We are confident that these explanations address the concerns raised and provide a more comprehensive view of the contributions and limitations of our work. We hope that these clarifications reassure the reviewer to revise their score to better reflect their positive opinion of our paper.

---

> > ### Comment · Reviewer_Bq7Y · 2024-08-10
> >
> > I thank the authors for addressing my comments. After reviewing the authors' response and the remaining concerns of other reviewers I believe this work would benefit from substantial revision and resubmission at a later date. I have changed by score to a borderline reject per my initial review as the authors have adequately acknowledge the particular typos mentioned here.
> >
> > The discussion of limitations provided in the rebuttal is interesting, and I agree with the feedback of other reviewers that this work is valuable; however, I think a more substantial revision which works a discussion of limitation and highlights the more particular contributions of this work would serve as a stronger submission in the future.

---

> > > ### Author Response · Authors · 2024-08-11
> > >
> > > We appreciate that the reviewer took the time to read our rebuttal and thank them for updating their score.
> > >
> > > However, we have to admit we are not quite sure why the reviewer still believes the paper requires substantial revision. From their review, we had understood that clarifying our contributions (as we did in the rebuttal) at the end of the introduction, adding a discussion of the wide-sense stationarity in Section 2 and the discussion of the limitations of our work in a new paragraph of the conclusion would sufficiently address the concerns raised by the reviewer. Can the reviewer help us understand their viewpoint by clarifying what is missing in our rebuttal? We appreciate further feedback so that we can continue improving our paper.

---

> > > > ### Comment · Reviewer_Bq7Y · 2024-08-13
> > > >
> > > > Thank you for the response.
> > > > Given the lack of substantial discussion in the original submission and the extent of the rebuttal provided, I still have some concerns regarding the ability to fit this additional material within the revision period. Given the feedback provided to other reviewers and since Neurips does not support revision uploads, I will have to take the authors at their word that their revisions and the inclusion of the material from the rebuttal will not affect the clarity and organization of the revised manuscript.
> > > >
> > > > I will raise my score to a borderline accept in light of this consideration, and wish you luck for the final decision.
> > > >
> > > > To the Neurips area chairs and board - I would encourage in the future allowing authors to submit revised manuscripts during the rebuttal period so that reviewers can guarantee that issues of clarity and promised changes to the manuscript are actually delivered.
> > > >
> > > > To the authors and other reviewers, I have rarely seen an active discussion during this period between authors and reviewers and I was happy to see the substantial time and effort put into revising this work on both sides.

---

> > > > > ### Author Response · Authors · 2024-08-13
> > > > >
> > > > > We thank the reviewer for their confidence and for increasing their score.
> > > > >
> > > > > We join the reviewer in thanking all the reviewers for the active discussion and in helping us sharpen the contributions of our work.

---

### Official Review · Reviewer_Bgq8 · 2024-07-11

**Soundness:** 3
**Presentation:** 3
**Contribution:** 4
**Rating:** 7
**Confidence:** 4

**Summary:**

The paper explores challenges RNNs face in learning long-term dependencies. While generally attributed to the exploding and vanishing gradients problem (EVGP), the authors reveal that as a network's memory increases, parameter changes cause an explosion of the second moment of hidden states and their gradients, making gradient-based learning highly ill-posed. They study different design choices in recurrent architectures such as element-wise recurrence, gating mechanisms and careful parameterization with respect to the curse of memory. They apply their theoretical findings to successful architectures such as state-space models (SSMs) and LSTMs, offering new insights into why some RNN architectures perform better in gradient-based learning.

**Strengths:**

- The paper addresses a fundamental problem, the curse of memory, and connects it through theoretical analysis and experimental validation to the success of recent architectures, which makes the insights incredibly valuable to the machine learning community
- I like how the authors connect the dots between multiple facets of model training, e.g. investigating the implications of RNN recurrence parametrization on adaptive LR optimizers, which are actually used in practice
- The paper is also well structured and generally well written.

**Weaknesses:**

- To add to EVGP mitigation techniques (l. 61 - 64): There also exist regularization techniques based on dynamical systems theory, see especially [1]
- There are quite some typos across the document, see the list below. I recommend running the paper through some sort of language checker to improve the manuscript from a readability and presentation side.
- The authors repeatedly call $\mathbb{E}[X^2]$ a variance (where $X$ here is a placeholder for any RV, i.e.  $h_t$ and $d_\theta h_t$ in the paper), which is technically not correct. It is the second moment. I’d consider fixing this or explaining why the missing term for the variance, $\mathbb{E}[X]^2$ (first moment squared), can be neglected.

Typos / text-bugs:
- l. 119: “[...] a low a pass filtered [...]
- l. 191: “[...] does not hurts [...]”
- l. 192 “Several RNN architectures implicitly alleviates [...]”
- l. 200 “While such models can can approximate any smooth mappings [...]”
- l. 219 “anaylsis”
- l. 257 “Both network are trained [...]”
- word missing in l. 283 “[...] that characteristic of the curse of memory, [...]”?
- l. 298 consistently -> consistent

[1] Schmidt et al., Identifying nonlinear dynamical systems with multiple time scales and long-range dependencies (ICML 2021)

**Questions:**

- in l. 180 you assume $\gamma$ is independent of $\lambda$ yet in the Diff. eq. in l. 182 you introduce the dependency again ($\gamma(\lambda)$)? Is this connected to what you write in the caption of Fig. 2 where you decouple $\gamma$ from $\lambda$ when differentiating? Can you explain the reason for this in more detail?
- in l. 289 you mention that you probe ADAM’s effective learning rate by providing a vector of ones to the optimizer - does that mean you simply read out ADAM’s (corrected) second-moment estimates and report $\frac{\eta}{\sqrt{v_n} + \epsilon}$, where $\eta$ is the global learning rate and $v_n$ the second moment estimates at iteration $n$? If so, at which iteration/epoch do you query the effective learning rate?

**Limitations:**

While an explicit section on limitations is missing, the authors mention their limitations and specific assumptions throughout the manuscript.

---

> ### Author Rebuttal · Authors · 2024-08-06
>
> We appreciate the reviewer’s very positive feedback and their valuable input regarding these additional references and important details. We address their questions below.
>
> > in l. 180 you assume $\gamma$ is independent of $\lambda$ yet in the Diff. eq. in l. 182 you introduce the dependency again ($\gamma(\lambda)$)? Is this connected to what you write in the caption of Fig. 2 where you decouple from $\lambda when differentiating? Can you explain the reason for this in more detail?
>
> Indeed, $\gamma$ must be a function of $\lambda$ for the normalization to be effective. However, including $\lambda$ inside the normalization significantly complicates our analytical formula, making it harder to reason about them in the backward pass. This is why, for that section, we reason about $\gamma(\mathrm{stopgradient}(\lambda))$.
>
> > does that mean you simply read out ADAM’s (corrected) second-moment estimates?
>
> The quantity we report incorporates both the first-moment and second-moment estimates from Adam, not just the second-moment one.
>
> > at which iteration/epoch do you query the effective learning rate?
> In the plots presented, we query the effective learning rate at the very end of training, using a constant learning rate (in contrast to the cosine scheduler used in our other experiments). However, our experimental pipeline monitors the effective learning rate at every epoch, and we have observed that the trends remain consistent throughout training.
>
> We recommend the reviewer to take a look at the new experiments we report in the global answer. They show that our theory extends to the gated case and to a larger class of practically relevant architectures. We hope the reviewer may take these additional results into account and consider adjusting their assessment accordingly.

---

> > ### Comment · Reviewer_Bgq8 · 2024-08-12
> > **Re: Rebuttal**
> >
> > I thank the authors for the clarifications, elaborate rebuttal and additional experiments in the general response. I think this work is of great value for the community. Hence, I increased my score to 7.

---

> > > ### Author Response · Authors · 2024-08-13
> > >
> > > We thank the reviewer for their positive feedback on our work and for increasing their score!

---

### Official Review · Reviewer_uPdc · 2024-07-12

**Soundness:** 3
**Presentation:** 2
**Contribution:** 2
**Rating:** 4
**Confidence:** 3

**Summary:**

The paper discusses "the curse of memory" which is the hypersensitivity of hidden states to parameters as the memory increases. This can lead to optimization issues, even if the exploding/vanishing gradient issue is addressed. The authors discuss solutions to this problem: complex diagonalization, normalization, and reparametrization. To this end, the authors derive closed-form solutions for the variance of the hidden state and its sensitivity to the parameters and discuss how the solutions can prevent the variances from diverging. This can give insights into why certain families of RNNs are performant. In particular, they show that both the variance of the hidden state and its derivative wrt to the parameters diverges as the memory goes to infinity (although at different rates). Therefore, normalizing the hidden state and reparametrizing the variable representing the memory's temporal scale can avoid the divergence issue.

**Strengths:**

- The paper puts forward closed-form solutions for the variance of hidden state and the sensitivity of hidden state to parameters. This is interesting and can provide insights into RNN dynamics and possibly network design.
- The results and the discussion on adaptive learning rate and how it relates to the Hessian of fully connected and complex diagonal RNNs are interesting and, to my knowledge, novel.

**Weaknesses:**

- The main weakness of the paper is the presentation. It is unclear what the contribution of papers is. At many points in the manuscript, it is unclear if the authors are discussing previous results on LRUs or putting forward new results. It would be good if the authors explicitly spell out their contributions.
- Although the paper is presented as discussing the curse of memory and the solutions to it, it reads more like "why LRUs are performant" The solution discussed: exponential reparametrization, normalization, and diagonalization, are all introduced and employed in LRU architecture [20]. However, the authors only discuss SSMs and gated RNNs as networks that implicitly address "the curse of memory". Meanwhile, LRUs employ the same solution discussed in the paper explicitly. Although LRUs can be thought of as a subclass of SSMs, vanilla SSMs [17] do not use the normalization, diagonalization, and exponential reparametrization discussed in the paper. Therefore, discussing LRUs only in experiments "to represent SSMs due to its simplicity" is not justified.
- If the solution to the curse of memory is not the contribution of the paper, then the novelty and contribution become limited. As I mentioned earlier, LRUs employ the same solution and sometimes with the same justification, for example, exponential parametrization is used for higher granularity around 1 [20].
- To motivate their results, the authors mention that "We answer negatively by showing that gradients can explode as the memory of the network increases, even when the dynamics of the network remains stable." but it is unclear where this is shown.

- minor comments:
L117: the paragraph on backward pass is confusing and hard to understand.
L139: check wording. Could not understand it.

**Questions:**

- What are the main contributions of the paper? Which results and which sections?
- What are other types of normalization and reparametrization, other than the ones employed by LRUs, that can address the curse of memory?

---

> ### Author Rebuttal · Authors · 2024-08-06
>
> We thank the reviewer for their review and address their concerns below.
>
> **On the contributions of the paper**
>
> > What are the main contributions of the paper?
>
> We provide a detailed list of our contributions in the global answer.
>
> > If the solution to the curse of memory is not the contribution of the paper, then the novelty and contribution become limited.
>
> In this paper, our objective is not to design a new architecture capable of avoiding the curse of memory — but instead to describe this issue thoroughly (it has never been reported before), and argue why recent architectures (e.g. SSMs) and classical ones (LSTMs, GRUs) have inner mechanisms alleviating the curse of memory. As we describe in the paper, the issue we study is deeply linked with the optimization of any recurrent model capturing long-range interactions, and translates into challenging loss landscape properties that we show can be alleviated by diagonality and reparametrization. Indeed, as the LRU paper (and others) shows, linear RNNs that are functionally equivalent (i.e. real dense linear RNNs, complex diagonal linear RNNs, reparametrized linear RNNs) have drastically different performances on tasks such as the ones within the long-range arena benchmark. Our paper dives deep into this intriguing finding and finds a direct effect of some of the commonly used tricks on the loss landscape.
>
> **On the different architectures**
>
> > What are other types of normalization and reparametrization, other than the ones employed by LRUs, that can address the curse of memory?
>
> We decided to focus on the LRU due to its simplicity, but it is far from being the only architecture capable of addressing the curse of memory. As we argue in Section 3, the discretization of a continuous-time ODE used in SSMs (more detail on that in the next paragraph) and the input and forget gates of LSTMs / GRUs partly serve the same purpose.
>
> > Meanwhile, LRUs employ the same solution discussed in the paper explicitly.
>
> > vanilla SSMs [17] do not use the normalization, diagonalization, and exponential reparametrization discussed in the paper.
>
> > Therefore, discussing LRUs only in experiments "to represent SSMs due to its simplicity" is not justified.
>
> While the LRU paper discusses many of the techniques we study in our paper, their investigation is mainly empirical. The objective of our paper is instead to deepen our theoretical understanding of RNN architectures like LRUs, SSMs, as well as the classical LSTMs/GRUs. Our path is of course guided by the design of such architectures, but our findings do not simply revisit the relative papers: we go one step further and show how these models can partially solve the curse of memory issue through reparametrization, normalizations, and diagonal structure. The LRU is arguably the simplest model capable of alleviating the curse of memory. While we agree that the structure of SSMs such as S4 is quite different, it is already pointed out by the LRU authors themselves (see the last section in their paper), that many of the tricks employed in their design can be traced back to mechanisms used in S4: it uses a diagonal parametrization of the recurrence, delta serves as normalization, and ZOH discretization induces an exponential stable parametrization.
>
> **On the writing**
>
> > […] it is unclear where this is shown.
>
> This sentence refers both to our analytical and empirical results. In our analytical results, we show that $\mathrm{d}_\theta h_t$, and therefore the loss gradient, grows to infinity as $\lambda$ goes to $1$ (”the memory increases”). Figure 5.B shows that the gradient of the loss explodes in practice (there is a typo in the legend of this figure: the quantity reported here is the gradient of the loss). In all cases, the dynamics of the network are stable as $\lambda \in [0, 1)$.
>
> We are confident that our remarks address the reviewer's concerns and provide necessary clarification, and we remain available during the discussion period for any additional questions. We encourage the reviewer to examine the new results reported in our global answer, which substantially extend the scope of our work, and we hope that our rebuttal will prompt a favorable reassessment of our paper's contributions and impact.

---

> > ### Comment · Reviewer_uPdc · 2024-08-10
> >
> > I thank the authors for taking the time to respond to my comments and other reviewers comments. I will comment on the issues raised by myself here.
> >
> >  I appreciate the importance of the topic and the fact that the authors study the RNNs and LRUs from a more theoretical standpoint. My general concern is presentation and I believe that the paper will greatly benefit from major reorganization.
> >
> > While the general solution to the curse of the memory is discussed, the specific solutions, i.e., the specific form of reparameterrization and specific form of normalization are the forms employed by LRUs. But the LRUs are not really discussed, or mentioned, in the preceding sections. I would suggest to either make the paper “a theoretical study of LRUs” or discuss more specific solutions, e.g., S4, and make it more general. The list of the contributions stated in the global rebuttal is a good starting point. I encourage the authors to resubmit the manuscript after reorganization and revision.  Therefore, I will keep my rating as “Technically solid paper where reasons to reject outweigh reasons to accept.”

---

> > > ### Author Response · Authors · 2024-08-11
> > >
> > > We thank the reviewer for their time and valuable input. Based on our understanding, the reviewer is suggesting two major revisions: first, make the paper less LRU-centric, and second, use the contribution statement from our rebuttal to reorganize the paper.
> > >
> > > Regarding the first point, we agree that this criticism applies to the empirical part of the submitted paper, but we believe that Figure 3 from the PDF already addresses this concern. We additionally believe that focusing on one architecture in most of the paper, while discussing how our findings apply to other architecture (both theoretically and empirically), helps us in keeping the exposition simple.
> > >
> > > As for the second point, our rebuttal’s contribution list follows the exact same structure the paper.
> > >
> > > As a consequence, we are uncertain how to structure the paper differently. Could the reviewer provide more specific feedback on what we could improve or clarify what additional experiments/results they would like to see in our paper?

---

> > > > ### Comment · Reviewer_uPdc · 2024-08-14
> > > >
> > > > I want to thank the authors and other reviewers for their active discussion. To clarify my first suggestion, I think the paper is LRU-specific both in theoretical section and emperical section. In the theoretical section, the form of the normalization and reparameteraztion discussed are the exact form used in LRUs. But LRUs are not mentioned at all until the experriment. Therefore, my suggestion was to EITHER
> > > > - make the paper *more LRU centric* from the beginning and make the paper about LRUs OR
> > > > - make the paper *less LRU centric* by discussing more forms of normalization and reparameterizations and including more architectures in the experiments.
> > > >
> > > > Regarding the contribution list, I suggest to include it in the paper. In the current form, it is not clear what the contribution of the paper is. Including the list provided in the global rebuttal in the manuscript will help the reader understand the paper contributions.

---

### Official Review · Reviewer_zt6Z · 2024-07-12

**Soundness:** 3
**Presentation:** 3
**Contribution:** 3
**Rating:** 5
**Confidence:** 4

**Summary:**

The paper provides a theoretical and practical analysis on the difficulties of training recurrent neural networks. In particular, given the current rise of interest in leveraging recurrent mechanisms for long sequence processing -- due to novel architectural components and solutions (deep state-space models, linear/diagonal recurrences, exponential parametrizations), identifying the crucial components allowing to preserve long range information is a very important research avenue.

**Strengths:**

The paper provides an interesting theoretical analysis that highlights which are the key components allowing modern solutions (Deep SSMs) and gated models (LSTMs/GRU) to achieve good performances. Moreover, it provides some proof-of-concept practical results that help to confirm the analysis. The paper is well written (apart from few typos, I suggest a proofreading).

**Weaknesses:**

I have some concerns with some of the paper initial framing and setting (see the Questions section) and the experimental analysis, which has been carried on very synthetic tasks which I am not sure are capable to represent real learning scenarios, thus possibly hindering the analysis validity.

**Questions:**

1) (Minor) While I understand that recent literature has completely overriden this notion, the term "state-space model" refers [1,2,3] to a very general broad concept which is common in control theory and dynamical systems -- which simply refers to a model equipped with state variables controlled by first-order differential equations or difference equations. Such definition  covers the typical feedback connection of Recurrent Models. Thus, stating in the abstract  and in the main text "state-space models, a subclass of RNNs" seem a bit too strong to me -- even if several recent works highlighted the connections. I would be completely fine with specifying "Deep" SSMs (just to highlight that the authors are referring to modern models) or "Structured" SSMs and to mention the suggested references, to better frame the paper to the readers.

2) In Eq. 1, the authors consider a model not equipped with an output mapping (that was instead considered in Eq. 10, i.e. y_t = Ch_t + Dx_t) -- the hidden state h_t is directly provided to the loss function.  Why is that the case, and do the considered analysis still hold when considering such mapping (that is common in RNNs and SSMs)?

3) The authors refers to an Infinite time horizon/infinite sequences (lines 98/100) as the basis for their analysis. Practically, how long the learning process necessitates to be to the considered issue to emerge/is this something that also the experimental analysis considers? In such case, does this have relation with recent literature highlighting the difficulties of sequential models when dealing with Infinite length sequences [3, 4]? I would like a comment by the authors on this.

4) Line (40): the notation (x_t)_t is not very clear to me.

5) Could the authors give an intuition on lines (113/116)?  Why an higher correlation in the input data should imply an higher variance in the model state (in my understanding, similar inputs imply similar loss function, thus smaller gradients)

6) In line 135, the authors use yet another form for the state update that do not consider the B linear mapping. Why is that the case?

7) The teacher-student task should be better described and justified. Why did the authors chose to tackle such task instead of other synthetic tasks (such as selective copy, induction heads etc. see recent surveys [3] for references on synthetic datasets ). And do the authors believe that the teacher-student task is general enough (i.e., points sampled from a normal distribution) to represent a setting where the state has to learn meaningful long range dependencies? Do the conclusions still hold when dealing with tasks  requiring to store more informative data? I believe that  experiments could benefit from experimentations on more "difficult" settings (selective copy, Long Range Arena) to confirm the theoretical considerations with real learning scenarios.

[1]  Genshiro Kitagawa. A self-organizing state-space model. Journal of the American Statistical Association, pages 1203–1215, 1998. 4

[2]  Anton Schafer and Hans Georg Zimmermann. Recurrent neural networks are universal approximators. In Proceedings of ICANN,  pages 632–640. Springer, 2006.

[3] Tiezzi, Matteo, et al. "State-Space Modeling in Long Sequence Processing: A Survey on Recurrence in the Transformer Era." arXiv preprint arXiv:2406.09062 (2024).

[4] Tiezzi, Matteo, et al. "On the resurgence of recurrent models for long sequences: Survey and research opportunities in the transformer era." arXiv preprint arXiv:2402.08132 (2024).

**Limitations:**

The paper does not discuss limitations.

---

> ### Author Rebuttal · Authors · 2024-08-06
>
> We thank the reviewer for their encouraging review. We answer their questions below:
>
> 1. **SSM terminology.** We agree with the reviewer that the SSM term has been overloaded. This is the reason why we try to use the RNN terminology as much as possible. That said, we appreciate the suggestion of the reviewer and will change the terminology to “deep SSMs”.
> 2. **Missing B, C and D?** We focus on the recurrence because the $B$, $C$ and $D$ mappings (from the usual SSM notations) are feedforward layers and signal propagation has been extensively studied in those layers (e.g. Glorot and Bengio 2010).
> 3. **Infinite sequences.** In our preliminary investigations, we found that having time sequences longer than 3 times the characteristic time constant of the network ($1 / 1 - \lambda$) is enough to consider the sequences infinitely long. The issue we mention appears from the very first parameter update and therefore is not directly related, to the best of our understanding, to the issues reported in the review (online learning and catastrophic forgetting).
> 4. Would changing it to $(x_t)$ make it clearer? We want to emphasize that this is a sequence of numbers here.
> 5. > Could the authors give an intuition on lines (113/116)?
>
> The point we are trying to make in these lines is the following: if we want to train a deep network, we need neural activity to stay of constant magnitude over the network depth (following the classical results on signal propagation in feedforward neural networks). It follows that a recurrent layer that blows up the magnitude of the input will hinder the learning of the rest of the network.
>
> > Why an higher correlation in the input data should imply an higher variance in the model state
>
> We do not have a proper intuition for that result. We would like to emphasize that correlation is here between time steps and not samples, which might break some intuition. Additionally, the intuition the reviewer has does not always apply: take a L2 loss, if the predictions are all the same (thus all similar) but extremely large, the gradients will also be large.
>
> 6. We do not use any $B$ mapping for the same reasons as described in 2.
>
> 7. **Justification for the teacher-student experiment.** That is indeed an important point that we didn’t comment on enough in the current version of the paper. We focused on this teacher-student task for 2 main reasons: First, we wanted to have an experiment in which we can easily control important details for signal propagation (e.g. magnitude of eigenvalues or the concentration of eigenvalues), which is almost impossible to do in other synthetic tasks. Second, we wanted to disambiguate signal propagation from other capabilities of architectures like memory. We believe that this setup is general enough for studying signal propagation as we can reproduce the same qualitative order between the architectures as in the LRU paper (on the LRA benchmark) and as our new analysis reveals that, up to a first approximation, inputs can be considered i.i.d. in our Wikipedia experiment (c.f. new results in the global answer).
>
> We believe that our rebuttal answers the questions of the reviewer and hope that the reviewer may update their score accordingly. We remain available during the discussion period to clarify additional points.

---

> > ### Comment · Reviewer_zt6Z · 2024-08-11
> >
> > I acknowledge that I have read the reviews and rebuttals and I thank the authors for taking their time to answer my questions, clarifiying some aspects of the work. I still have some concerns regarding questions 2 and 6 (the assumptions by the authors on analyzing simplified versions of the models -- do the results still hold when considering models equipped with B,C,D?), which I believe are important and require a more in depth justification.  I agree with other reviewers that the work is promising but requires some additional work to better organize the contribution (i.e., regarding my points: the aforementioned setting simplification on B,C,D mapping and the teacher-student experimental setting should be better framed in the context of the paper). Regardless, I will keep my positive score.

---

> > > ### Author Response · Authors · 2024-08-12
> > >
> > > We here provide additional details on the B, C and D mappings, in the hope that they will clarify the remaining concerns of the reviewer. These mappings operate token-wise whereas the recurrence mixes information between tokens. This fundamental difference justifies why B, C and D should be considered separately from the recurrence and similarly to the rest of the feedforward layers of the network. On the more technical side, integrating B and C in our theory only requires changing the quantities we give as input to our theory, e.g. we should consider the statistics of Bx instead of x ($C\delta$ instead of $\delta$ for the backward pass through the recurrence). Our theory does not aim to capture the role of $D$ given that it side steps the recurrence. To summarize, integrating these mapping does not affect our conclusions (Sections 4 and 5 confirm it empirically).
> > >
> > > We believe that the concerns previously raised by the reviewer (as well as the ones mentioned by other reviewers) only require minor modifications to the corresponding sections of the manuscript (see also our answers to reviewers uPdc and Bq7Y). We would appreciate additional input from the reviewer to help us better identify which parts require substantial modifications before the next version of the paper.

---

### Author Rebuttal · Authors · 2024-08-06

We are grateful to the reviewers for their high-quality feedback and their positive comments about our paper. We answer the main points below and respond to each reviewer more specifically in corresponding threads.

**Contributions-limitations.** Many reviewers highlighted that the contributions and the limitations of the paper should be made clearer. We list them below:

- **Contributions**
    - We identify, **for the first time**, a crucial issue arising in the training of recurrent neural networks. We study it analytically in great detail in a simplified, yet realistic, setting. We then highlight that **successful RNN architectures** all share a common feature: they **mitigate this issue**.
    - We confirm that our theory accurately **captures learning dynamics in a controlled teacher-student regime**. We additionally show how our analysis enables understanding parts of the Hessian structure, thereby improving our understanding of the critical differences between parametrizations/architectures and how they **interplay with different optimizers**.
    - Overall, our paper brings theoretical insights into the training of recurrent neural networks that are of **great practical relevance**. Those results are particularly important as the interest in RNNs has been rising in recent years and there is **still very little literature** on the topic.
    - Reviewers uPdc and xzKN rightly pointed out that we should better position our paper in comparison to the LRU paper. The **key distinction lies in the question we study**: the LRU paper aims to extract **which** components are critical to the great performance of SSMs, whereas we aim to understand **why** (some of) these components are critical. Their paper is therefore mostly empirical while ours is theoretical. That said, our work **naturally builds on the LRU**: First, we found their architecture to be more amenable to theoretical analysis than SSMs. Second, we designed our teacher setup task to reproduce the qualitative findings of the LRU paper on a more sophisticated task (a discussion of this point in Sec. 4 is missing). Third, our Eq. 5 generalizes their Eq. 6 to more general input distributions.
- **Limitations**
    - Our theoretical analysis makes three main assumptions that are reasonable, yet have some limitations.
        1. The recurrence is **diagonal and time-invariant**. Our analysis **does not exactly capture signal propagation in the gated case**. However, we provide new results (see below + PDF) demonstrating that our theory is a good proxy for gated RNNs at initialization.
        2. The sequences are **infinitely long.** In practical terms, our results only **apply whenever the sequence length is larger than the largest characteristic timescale of the network** (in our preliminary investigations we found x3 to be large enough).
        3. Input sequences are **wide-sense stationary.** This is a very **standard assumption** in the analysis of linear time-invariant filters. Yet, many real-world processes are not stationary, e.g. the Wikipedia data we are considering in Sec. 5. We provide new results (c.f. next paragraph) that show that our **analysis is still accurate enough** when this assumption is not met.
    - The technical tools we use for our theoretical analysis are rather elementary and there is little to no hope that they generalize to highly nonlinear dynamics.
    - Signal propagation that is independent of the time scale considered is necessary for efficient gradient-based learning but not sufficient. There are many questions that we thus cannot answer with such an analysis. For example, our theory cannot give any insight into how much memory a model can store, or tell us how learned networks will generalize to longer sequences than the training ones.

**New results.** Following the reviewers’ questions regarding 1) whether our analysis holds when the inputs are not wide-sense stationary and 2) how much it holds for gated RNNs, we provide a refined empirical analysis of signal propagation in the setting of Sec. 5.

1. The Wikipedia dataset is not wide-sense stationary and of finite length so we cannot directly apply the results of our analysis. Instead, we compute the empirical auto-correlation function (PDF Fig. 1). We find that we can approximate it well with one i.i.d. component ($\rho = 0$) and a slowly decaying one ($\rho \approx 1$). This is the autocorrelation function we plug into our analytical formulas when needed.
2. In Sec. 3, we argued that gated RNNs can behave like the diagonal networks we study theoretically. We verify this empirically in Fig. 2 of the PDF by looking at the diagonal and non-diagonal components of the recurrent Jacobian $\mathrm{d}h_t/ \mathrm{d}h_0$ of a GRU. We find that the non-diagonal components are indeed negligible in standard initialization schemes (this breaks when we multiply the hidden to hidden connections by a factor ~3).
3. We study signal propagation in a simplified GRU network (same as in Chen et al. 2018) and compare it with our theory (slightly modified to take into account normalization). We find that our theory almost perfectly captures the time-invariant case and adequately captures the gated case.

We are confident that our rebuttal addresses the concerns raised by the reviewers and hope that they will consequently update their scores. Given that we cannot update the paper, we have integrated those remarks, as well as fixed typos, in what will be the next version of the paper.

---

### Comment · Area_Chair_dD58 · 2024-08-10

Dear Reviewers,

The authors have provided a comprehensive rebuttal. I encourage all of you to respond before the deadline, to have sufficient time to discuss outstanding issues. Please change your scores accordingly, with a small motivation based on the points that were addressed or not (especially concerning the relation with the LRU paper, the novelty of the analysis, and the technical assumptions underneath the methods).

Thanks, the AC

---

### Decision · Program_Chairs · 2024-09-25

**Decision:**

Accept (poster)

**Comment:**

The paper analyses what they call the "curse of memory" in recurrent neural networks (a term borrowed from prior research), i.e., the fact that the variance of the activations can exponentially diverge with respect to the size of the memory for very long sequences. In a sense, the analysis is dual to the one over gradients, as both exploit long chain of Jacobians over the sequence time steps.

The paper had a very live rebuttal period, at the end of which it moved from a potential reject (3/4/4/5/6), to a scenario where one reviewer is recommending acceptance and all the other reviewers remain non-committal with borderline scores. Some parts of the discussion have been correctly summarized by the authors here on OpenReview. Before proceeding to the decision, I would like to weigh on some remaining concerns:

* Reviewer `zt6Z` is concerned that the analysis only involves the transition matrix of SSMs ($\mathbf{A}$ in common notation), and not a full-fledged model. In addition, they were concerned about the use of an artificial scenario to validate the results. From what I understand, both points were addressed but the reviewer is not convinced they can be integrated easily in a camera-ready version. After going through the paper carefully, I agree on this with the authors that both points are relatively minor modifications to the paper, and I respectfully disagree with the reviewer (who, in any case, voted for a borderline accept).

* Reviewer `uPdc` is concerned about the relation of the paper with the LRU paper by Orvieto et al. In their opinion, the paper should be made more "LRU-centric", and they are also concerned that "*LRUs are not mentioned at all until the experriment*". The authors have argued that this paper complements the LRU paper, since most of the techniques of LRU (e.g., input normalization) are introduced empirically, and they are motivated from a signal propagation point of view here. I note that the authors have been very careful with prior literature (e.g., page 3, "*Martens and Sutskever [40] hypothesized that such a phenomenon could arise in RNNs*"), and LRUs are indeed cited in Section 2.2a as an example of application of the paper. However, the analysis does hold for other types of architectures. Hence, after careful analysis, I side with the authors also on this point.

* Reviewer `Bq7Y` has a few concerns (e.g., typos, discussion of the limitations) which were solved in the rebuttal period. They are now in a position similar to `zt6Z` concerning a potential inability to fit the material for the camera ready. On this, I have the same opinion as for `zt6Z`, since most modifications appear to be small.

* Reviewer `xzKN` is concerned about the assumption of wide sense stationarity made by the authors. This does not invalidate their claim, but they propose a different condition that should be less restrictive. This does not appear to be especially critical with respect to the results, as the authors also argue.

Importantly, all reviewers agree on the novelty of the paper and the potential for further investigation for RNNs, both in the original reviews and during the rebuttal period. I believe this is a strong point in favor of the paper. In addition, my own evaluation is that the paper is very well written, easy to follow (despite being fundamentally a theoretical paper), well organized, and with clear and easy to read figures and experiments that complement the analysis. None of the modifications discussed above would modify this substantially, and I do not see a strong reason for waiting another review cycle for incorporating them. Hence, I suggest acceptance for NeurIPS due to the importance of the topic and the results.